# ParaSolver-Turbo: Accelerating Parallel Diffusion Integrator via Intrinsic Partially Linear Structure

## Abstract

This paper explores the challenge of accelerating the sequential inference process of Diffusion Probabilistic Models (DPMs). We tackle this critical issue from a dynamic system perspective, in which the inherent sequential nature is transformed into a parallel sampling process. Specifically, we first reveal that the sequential integral solver of the diffusion model can be approximated by a full linear solver, enabling efficient computation for parallel integral solvers of DPMs. Based on such a linear formulation, we then introduce a unified framework that reformulates the original nonlinear sequential integral process of diffusion model as a system of partial linear equations. Moreover, we further develop an immediate update strategy to solve the system. In addition, we prove that (1) the system admits a unique root corresponding precisely to the trajectory of the sequential integral solver; (2) solving the system guarantees convergence to the trajectory of sequential integral solvers in equal or fewer iterations. Building on these insights, we present *ParaSolver-Turbo*, a partial linear parallel integral solver to accelerate a broad class of sequential and parallel sampling methods such as DDPM and Para-Solver. Extensive experiments validate that ParaSolver-Turbo achieves $2\times \sim 50\times$ speedup in terms of wall-clock time without measurable quality degradation. The source code will be released publicly.

## 1 Introduction

In recent years, the rise of diffusion probabilistic models (DPMs) (Ho et al., 2020; Song et al., b) has dramatically transformed the field of generative modeling. These models have become a cornerstone technique for a wide range of applications (Yang et al., 2024; Liu et al., 2023a; Lu et al., 2024; Chung et al., 2023; Lu et al., 2023; You et al., 2025; Liu et al., 2025a), from high-fidelity image and video synthesis (Rombach et al., 2022; Blattmann et al., 2023) to molecular generation (Wu et al., 2024; Nguyen et al., 2023). At their core, diffusion models operate through an iterative denoising process, mathematically formulated as an ordinary or stochastic differential equation (ODE/SDE). This process gradually refines noise from an initial Gaussian distribution, transforming it into a realistic sample that aligns with the target data distribution. However, a key limitation of DPMs is their slow sampling speed. Generating high-quality outputs typically requires numerous sequential denoising steps, each involving computationally intensive evaluations of large neural networks.

To accelerate sampling, researchers have proposed various approaches (Gong et al., 2024; Zheng et al., 2023; Gonzalez et al., 2024; Geng et al., 2024; Liu et al., 2023b; Luo et al., 2023). One line of work focuses on advanced SDE/ODE solvers, such as DDIM (Song et al., a) and DPMSolver (Lu et al., 2022), which leverage mathematical optimizations to reduce sampling steps. However, these methods often trade off sample quality for speed. Another direction involves distilling the ODE trajectory of a pre-trained diffusion model into a faster neural network, as seen in (Salimans and Ho; Song et al., 2023; Lipman et al.; Liu et al.). While effective, such techniques frequently suffer from degraded output quality and diversity. To overcome these limitations, recent work has explored parallel sampling strategies. Shih et al. (2024b) introduced a parallel denoising method based on Picard iteration, while Tang et al. (2024) reformulated the problem as solving triangular nonlinear equations via fixed-point iteration. Building on these advances, ParaSolver (Lu et al., 2025) models the sampling process as a banded nonlinear system, exploiting sparsity for greater efficiency than prior dense-structured approaches like ParaTAA. These methods offer three key advantages: (1) they

are training-free and compatible with existing solvers; (2) they preserve sample quality comparable to sequential sampling; (3) they drastically reduce sampling steps, significantly boosting efficiency.

Despite their promise, current parallel methods face two major bottlenecks. First, each iteration requires substantially more neural function evaluations (NFEs) than sequential sampling, causing parallel efficiency to saturate quickly due to GPU compute limits. Second, while iteration counts are reduced compared to sequential sampling, they remain high to constrain maximum speedup. To push these boundaries further, we propose *ParaSolver-Turbo*, a unified framework that generalizes prior work through the lens of nonlinear equations (NEs). Unlike existing methods that rely solely on fully nonlinear systems, we reinterpret sequential sampling as a mixed system of linear and nonlinear equations. This partial linear structure drastically reduces NFE overhead, unlocking new levels of parallel efficiency. Moreover, our framework is highly general—prior methods like ParaDiGMS, ParaTAA, and ParaSolver emerge as special cases within our formulation.

In summary, this paper makes the following theoretical and practical contributions:
- we introduce ParaSolver-Turbo, a novel parallel sampling algorithm for diffusion models, which reformulates the sequential nonlinear diffusion sampling process into a system of partial linear equations;
- we enhance the updating process through an immediate updating strategy, with theoretical guarantees that convergence to the sequential integral solver's trajectory is achieved within equal or fewer iterations; and
- we experimentally demonstrate that ParaSolver-Turbo achieves a $2.7\times$–$50.0\times$ speedup in wall-clock time while maintaining comparable generation quality (measured by FID and CLIP scores), setting a new state-of-the-art in this domain.

## 2 RELATED WORK

Parallel sampling has become a promising technique to speed up the sequential sampling process in DPMs while maintaining the same sample fidelity. By simultaneously denoising multiple sample points along ODE/SDE trajectories, this method iteratively improves a set of initial trajectories, allowing faster generation.

To the best of our knowledge, only a handful of recent works (Shih et al., 2024a; Tang et al., 2024; Lu et al., 2025) have investigated parallel sampling for DPMs. Among these, ParaDiGMS (Shih et al., 2024a) stands as the first parallel sampling method, considering discretized sample points along ODE trajectories as a sequence of fixed points that are refined iteratively using the fixed-point theorem and Picard-Lindelöf theorem. ParaTAA (Tang et al., 2024) builds upon fixed-point iteration by formulating DPM sampling as solving triangular nonlinear equations, where all discretized sample points serve as unknown variables. ParaSolver (Lu et al., 2025) further progresses the field by modeling the sampling process as a banded nonlinear equation system. Its hierarchical parallel strategy exploits the system's sparsity, achieving considerably higher computational efficiency compared to ParaTAA's dense triangular structure.

Although these methods demonstrate promising benefits, their parallel efficiency is constrained by two critical factors. First, they require substantially more NFEs per iteration than the sequential sampling, causing parallel efficiency to plateau quickly as NFEs scale—the single GPU's floating-point operations per second (FLOPS) become the bottleneck. Second, despite achieving a notable reduction in iterations relative to sequential approaches, the remaining iterations are still high, capping the maximum achievable parallel speedup.

In contrast, we reinterpret the sequential sampling of DPMs as a mixed system of linear and non-linear equations—a distinct departure from existing methods that require fully nonlinear equations. This partial linear structure substantially cuts down the number of NFEs required by current parallel methods, thereby significantly unleashing the parallel efficiency. In addition, our parallel framework is highly general; existing methods such as ParaDiGMS, ParaTAA, and ParaSolver emerge as special cases within our unified paradigm.

Beyond parallel sampling, we also observe other research directions focused on accelerating diffusion models, including model/sample partitioning (Ma et al., 2024b;a; Wang et al., 2024; Li et al., 2024; Liu et al., 2025b), gradient guidance (Wang et al., 2025), trajectory stitching (Pan et al., 2024), nested diffusion (Elata et al., 2024), minimum denoising step prediction (Yu and Barad, 2024), picard consistency model (So et al., 2025), and theoretical guarantees (Gupta et al., 2024; Chen et al.,

2024; Zhou and Sugiyama, 2024). Our work complements these approaches by parallelizing DPMs' inherently sequential sampling procedure.

## 3 PRELIMINARY

**Sequential Integral Solvers**. The diffusion dynamics are governed by a drift coefficient function $f(t)$ and a diffusion coefficient function $g(t)$. The forward diffusion process, which gradually transforms a clean image $\mathbf{X}_T$ into random noise $\mathbf{X}_0$ (note the convention where clean images correspond to $t = T$ and noise to $t = 0$), is described by the following stochastic differential equation (SDE):

$$d\mathbf{X}_t = f(t)\mathbf{X}_t dt + g(t)d\mathbf{W}, \tag{1}$$

where $d\mathbf{W}$ indicates the standard Wiener process. The generation process requires reversing this diffusion, leading to the reverse-time SDE:

$$d\mathbf{X}_t = \underbrace{\left[ f(t)\mathbf{X}_t - g^2(t)\nabla_{\mathbf{X}_t} \log p(\mathbf{X}_t) \right]}_{\varphi(\mathbf{X}_t, t)} dt + \underbrace{g(t)}_{\sigma_t} d\mathbf{W}, \tag{2}$$

where $\nabla_{\mathbf{X}_t} \log p(\mathbf{X}_t)$ is approximated by a neural network-based score function $\mathbf{S}_\theta(\cdot)$ with parameters $\theta$. The terms $\varphi(\mathbf{X}_t, t)$ and $\sigma_t$ parameterize the deterministic and stochastic processes for the reverse process, respectively. Let $\Phi(t, s, \mathbf{X}_s)$ denote the integral solution of Eq. (2) over the time interval $[s, t]$ with initial condition $\mathbf{X}_s$:

$$\Phi(t, s, \mathbf{X}_s) = \mathbf{X}_s + \int_s^t \varphi(\mathbf{X}_\tau, \tau)d\tau + \int_s^t \sigma_\tau d\mathbf{W}. \tag{3}$$

**Parallel Integral Solvers**. The ParaSolver framework (Lu et al., 2025) provides a unified approach that encompasses both parallel and sequential sampling methods. Here we introduce existing parallel algorithms within this framework. Consider an array of ODE/SDE trajectories $\{\mathbf{X}_t\}_{t=0}^T$ for DPMs[1]. ParaSolver (Lu et al., 2025) proposes a hierarchical parallel strategy that first divides the complete trajectory into $N$ sub-intervals $[t_0, t_1, \ldots, t_N]$ with $0 = t_0 < t_1 < \cdots < t_N = T$, then formulates a system of banded nonlinear equations:

$$\begin{cases} \hat{\mathbf{X}}_{t_0} - \mathbf{X}_{t_0} = \mathbf{0}, \\ \hat{\mathbf{X}}_{t_{n+1}} - \Phi(t_{n+1}, t_n, \hat{\mathbf{X}}_{t_n}) = \mathbf{0}, & n \in \{0, 1, \cdots, N-1\}, \end{cases} \tag{4}$$

where $\Phi(t_{n+1}, t_n, \hat{\mathbf{X}}_{t_n})$ represents the local integration operator between consecutive time steps. The framework generalizes various approaches, becoming equivalent to sequential sampling when $N = 1$ and matching full parallel methods like ParaDiGMS (Shih et al., 2024a) and ParaTAA (Tang et al., 2024) when $N = T$.

The algorithm proceeds through iterative refinement of an initial trajectory estimate $\{\hat{\mathbf{X}}_{t_i}\}_{i=0}^N$, typically initialized with coarse approximations or random noise, denoted as $\left\{ \hat{\mathbf{X}}_{t_i}^{(0)}, i = 0, \cdots, N \right\}$. Let $\hat{\mathbf{X}}_{t_0:t_N} = [\hat{\mathbf{X}}_{t_0}^\top, \ldots, \hat{\mathbf{X}}_{t_N}^\top]^\top$ denote the concatenated state vector. Then, at the $k$-th parallel iteration ($k \in [0, K]$), the estimated trajectory is updated according to:

$$\hat{\mathbf{X}}_{t_0:t_N}^{(k+1)} = \hat{\mathbf{X}}_{t_0:t_N}^{(k)} - \mathcal{G}^{(k)}\mathcal{R}_{t_0:t_N}^{(k)}, \tag{5}$$

where $\mathcal{R}_{t_{n+1}}^{(k)} = \hat{\mathbf{X}}_{t_{n+1}}^{(k)} - \Phi(t_{n+1}, t_n, \hat{\mathbf{X}}_{t_n}^{(k)})$ and $\mathcal{R}_{t_0}^{(k)} = \hat{\mathbf{X}}_{t_0} - \mathbf{X}_{t_0} = \mathbf{0}$ indicate residual terms to be minimized. The matrix $\mathcal{G}^{(k)}$ distinguishes various algorithms for solving the above system of nonlinear equations: $\mathcal{G}^{(k)} = \mathbf{I}$ yields fixed-point iteration (Burden and Faires, 1985), while $\mathcal{G}^{(k)} = (\mathcal{J}^{(k)})^{-1}$ corresponds to the Newton-Raphson iteration (Kelley, 2003), with $\mathcal{J}^{(k)} = \partial\mathcal{R}_{t_0:t_N}^{(k)}/\partial\hat{\mathbf{X}}_{t_0:t_N}$ being the Jacobian matrix.

---

[1]For compatibility with parallel sampling, we discretize the continuous time interval $[0, T]$ into discrete steps $[0, 1, \ldots, T]$ with unit step size.

## 4 PROPOSED METHOD

Parallel solvers reformulate the sequential integration process of the diffusion model into a system of non-linear equations (as outlined in Eq. (4) and Eq. (5)). However, parallel solvers demand significantly more model evaluations per iteration to denoise multiple noisy samples compared to sequential solvers, substantially limiting their acceleration performance. To tackle this problem, we propose a linearization method, which reduces the fully nonlinear computations (i.e., the model evaluation operation) of parallel solvers to partially linear operations.

### 4.1 $\varrho$-NONLINEAR EQUATION SYSTEM

Existing approaches typically develop parallel solvers based on the sequential integral formulation in Eq. (3), which results in a fully nonlinear system of equations in Eq. (4) when employing the nonlinear score function $\mathbf{S}_\theta(\cdot)$. We reveal that the sequential integral solver in Eq. (3) can be expressed in a more elementary form and reduced to a linear solver, as formally stated in the following Proposition 1.

**Proposition 1** (Linear Approximation of the Sequential Solver). *The sequential integral solver from* Eq. (3) *can be decomposed into an linear solver $\mathcal{H}(\cdot)$ with a negligible error term $\mathcal{Z}$:*

$$\Phi(t, s, \mathbf{X}_s) = \mathcal{H}\left(t, s, \mathbf{X}_s, \mathbf{X}_T(\mathbf{X}_s)\right) + \mathcal{Z}, \tag{6}$$

*where the linear sequential integral solver $\mathcal{H}(\cdot)$ is linear with respect to $\mathbf{X}_s$ and $\mathbf{X}_T(\mathbf{X}_s)$:*

$$\mathcal{H}(t, s, \mathbf{X}_s, \mathbf{X}_T(\mathbf{X}_s)) = \frac{\alpha_t}{\alpha_s}\mathbf{X}_s - \alpha_t \int_{\lambda_s}^{\lambda_t} e^{-\lambda}\frac{\mathbf{X}_\lambda - \alpha_\lambda \mathbf{X}_T(\mathbf{X}_s)}{\sigma_\lambda}d\lambda + \int_{\lambda_s}^{\lambda_t} g(\lambda)d\mathbf{W}, \tag{7}$$

*and the error term $\mathcal{Z}$ satisfies $||\mathcal{Z}|| \leqslant \zeta \sigma_t \left(\frac{\alpha_t \sigma_s}{\alpha_s \sigma_t} - 1\right) \simeq 0$.*

*Proof.* See Appendix H for the complete derivation. $\qquad\square$

**Remark 1.** *Proposition 1 reveals that the exact sequential solver in* Eq. (3) *possesses an intrinsically simpler linear structure.*

Now, following ParaSolver (Lu et al., 2025), the linear sequential integral solver in Eq. (7) can be decomposed into many integral functions on different time subintervals, written as

$$\mathbf{X}_{t_{n+1}} = \mathcal{H}(t_{n+1}, t_n, \mathbf{X}_{t_n}, \mathbf{X}_{t_N}), n \in \{0, 1, \cdots, N-1\}. \tag{8}$$

Motivated by recent parallel algorithms (Lu et al., 2025; Song et al., 2021), this inherently sequential problem can be addressed in a parallel framework. Specifically, such a sequence of cascaded linear functions can be interpreted as a system of fully linear equations. Denote by $\{\mathbf{X}_{t_n}, n = 0, \cdots, N\}$ a set of points exactly on the diffusion trajectory, and $\left\{\hat{\mathbf{X}}_{t_n}, n = 0, \cdots, N\right\}$ the unknown variables to be optimized towards the trajectory. The cascade of integral formulations as Eq. (8) can then be reformulated as the solutions to the following *fully linear equations*.

**Definition 1** (Fully Linear Equation System). *We define the system of fully linear equations for the sequential sampling process in* Eq. (8) *as*

$$\begin{cases} \hat{\mathbf{X}}_{t_0} - \mathbf{X}_{t_0} = \mathbf{0}, \\ \hat{\mathbf{X}}_{t_{n+1}} - \mathcal{H}(t_{n+1}, t_n, \hat{\mathbf{X}}_{t_n}, \hat{\mathbf{X}}_{t_N}) = \mathbf{0}, n \in \{0, 1, \cdots, N-1\}. \end{cases} \tag{9}$$

The complete linearity of the above system permits exceptionally efficient solving with negligible computational cost. However, this very full linearity prevents them from achieving the sample quality of the sequential integral solvers in Eq. (3), as they lack information regarding the target data distribution. To bridge this quality gap, we introduce the target distribution information into the linear system by supplying estimated clean sample $\hat{\mathbf{X}}_{t_N}$. Recall that in the forward process of DPMs, we can compute a predicted clean sample $\hat{\mathbf{X}}_{t_N}$ as follows:

$$\hat{\mathbf{X}}_{t_N}(\hat{\mathbf{X}}_{t_n}, \theta) = \frac{\hat{\mathbf{X}}_{t_n} - \sigma_{t_n}\mathbf{S}_\theta(\hat{\mathbf{X}}_{t_n}, t_n)}{\alpha_{t_n}}. \tag{10}$$

where $\mathbf{X}_{t_n} \sim \mathcal{N}(\alpha_{t_n}\mathbf{X}_{t_N}, \sigma_{t_n}\mathbf{I})$ where $\alpha$ and $\sigma$ are the noise schedule. Note that the diffusion model consumes most computational resources on the score-matching process, i.e., estimating corresponding noise or clean samples by a neural network. In this paper, to lessen such burden of score matching, we explore the consistency of reverse diffusion process and propose to divide the NEs to a partial-linear equation system, namely $\varrho$-nonlinear equation system. Specifically, due to the diffusion models make a progressive transition from pure noise to clean data distribution, the score function is highly unstable on the high noise region, which is also evidenced by recent research (Karras et al., 2022; Ma et al., 2024b). Based on such an insight, to further balance the score estimation accuracy and computational consumption, we divide the original NEs shown as Eq. 4 into $\varrho$-Nonlinear Equation System in Definition 2, where for the unstable and stable score region, we apply the nonlinear and linear solvers, respectively.

**Definition 2** ($\varrho$-Nonlinear Equation System). *We define a $\varrho$-nonlinear equation system for the sequential sampling process in* Eq. (8) *as*

$$\begin{cases} \hat{\mathbf{X}}_{t_0} - \mathbf{X}_{t_0} = \mathbf{0}, \\ \hat{\mathbf{X}}_{t_{n+1}} - \mathcal{H}\Big(t_{n+1}, t_n, \hat{\mathbf{X}}_{t_n}, \hat{\mathbf{X}}_{t_N}(\hat{\mathbf{X}}_{t_n}, \theta)\Big) = \mathbf{0}, n \in \{0, 1, \cdots, \varrho - 1\}, \\ \hat{\mathbf{X}}_{t_{n+1}} - \mathcal{H}\Big(t_{n+1}, t_n, \hat{\mathbf{X}}_{t_n}, \hat{\mathbf{X}}_{t_N}(\hat{\mathbf{X}}_{t_{\varrho-1}}, \theta)\Big) = \mathbf{0}, n \in \{\varrho, \varrho + 1, \cdots, N - 1\}. \end{cases} \tag{11}$$

**Remark 2.** *The scalar $\varrho$ serves as a critical transition point between nonlinear (score) and linear (fixed score) regimes in the system. The first $\varrho$ equations evolve into computationally costly nonlinear equations, reducing into exiting nonlinear parallel system in Eq. (4). Subsequent equations (for $n \geqslant \varrho$) retain computational efficiency as linear equations, as they depend only on the fixed prediction from the $(\varrho - 1)^{th}$ time step.*

**Remark 3.** *The above framework establishes a unified paradigm, subsuming all existing parallel methods such as ParaDiGMS, ParaTAA, and ParaSolver as special cases. In particular, existing methods are fully nonlinear equation systems, corresponding to the limiting case $\varrho = N - 1$.*

**Remark 4.** *Our theoretical analysis demonstrates that the root to the $\varrho$-nonlinear equation system exists uniquely and provides an unbiased estimator for the sampling results from the sequential integral solver, as established in Proposition 2 and Proposition 3, respectively.*

To theoretically validate the equivalence, we define the cumulative state error as $\mathbf{\Delta}_{t_n} = \hat{\mathbf{X}}_{t_n} - \mathbf{X}_{t_n}$, representing the deviation between the parallel solution and the precise sequential trajectory. Additionally, based on Definition 2, the the clean sample estimator is explicitly defined as:

$$\mathbf{E}_{t_n} = \begin{cases} \hat{\mathbf{X}}_{t_N}(\hat{\mathbf{X}}_{t_n}, \theta) & \text{if } n < \varrho, \\ \hat{\mathbf{X}}_{t_N}(\hat{\mathbf{X}}_{t_{\varrho-1}}, \theta) & \text{if } n \geqslant \varrho. \end{cases} \tag{12}$$

Moreover, let $\boldsymbol{\xi}_n = \mathbf{E}_{t_n} - \mathbf{X}_T$ denote the clean sample estimation error.

**Proposition 2** (Equivalence under Non-Ideal Conditions). *For any $\varrho \in \{1, \ldots, N\}$, the solution to the $\varrho$-nonlinear equation system defined in* Eq. (11) *is equivalent to the true sequential integral trajectory $\{\mathbf{X}_{t_n}\}_{n=0}^{N}$. Specifically, the discrepancy is governed by the following error recurrence bound:*

$$\|\mathbf{\Delta}_{t_{n+1}}\| \leqslant |\mathcal{A}_n|\|\mathbf{\Delta}_{t_n}\| + |\mathcal{B}_n|\|\boldsymbol{\xi}_n\| + \|\mathcal{Z}_n\|, \tag{13}$$

*where $\mathcal{A}_n = \frac{\alpha_{t_{n+1}}}{\alpha_{t_n}} \log \frac{\alpha_{t_n}\sigma_{t_{n+1}}}{\alpha_{t_{n+1}}\sigma_{t_n}} \approx 0$ and $\mathcal{B}_n = \alpha_{t_{n+1}} \log \frac{\alpha_{t_{n+1}}\sigma_{t_n}}{\alpha_{t_n}\sigma_{t_{n+1}}} \cdot \log\left(\sqrt{\frac{\alpha_{t_n}\sigma_{t_{n+1}}}{\alpha_{t_{n+1}}\sigma_{t_n}}}\right) \approx 0$. Since $\|\mathbf{\Delta}_{t_0} = 0\|$, the coefficients $\mathcal{A}_n$ and $\mathcal{B}_n$ approach negligible values ($\approx 0$), and combined with $\|\mathcal{Z}_n\| \approx 0$, the total error $\|\mathbf{\Delta}_{t_{n+1}}\|$ remains negligible throughout the sampling process.*

*Proof.* See Appendix I for the complete derivation. $\square$

**Remark 5.** *Prop. 2 guarantees that different $\varrho$ does not impact the quality of the generated samples. Our experiments also confirm this, showing that varying $\varrho$ does not degrade the FID and CLIP scores of the generated sample across various diffusion processes like variance preserving (VP) and variance exploding (VE). We note that $\varrho$ is less of a hyperparameter and more of a hardware configuration parameter. For good efficiency, $\varrho$ can simply be set to the number of available processors.*

## 5 SOLVING THE $\varrho$-NONLINEAR EQUATION SYSTEM

We investigate the update rule in Eq. (5) to solve the prescribed $\varrho$-nonlinear system. Specializing Eq. (5) to our system yields the following iterative scheme:

$$\hat{\mathbf{X}}_{t_0:t_N}^{(k+1)} = \hat{\mathbf{X}}_{t_0:t_N}^{(k)} - \mathcal{G}^{(k)}\mathcal{R}_{t_0:t_N}^{(k)}, \tag{14}$$

where the residual term $\mathcal{R}_{t_{n+1}}^{(k)}$ with $\mathcal{R}_{t_0}^{(k)} = \hat{\mathbf{X}}_{t_0}^{(k)} - \mathbf{X}_{t_0} = 0$ is defined as:

$$\mathcal{R}_{t_{n+1}}^{(k)} = \begin{cases} \hat{\mathbf{X}}_{t_{n+1}}^{(k)} - \mathcal{H}\Big(t_{n+1}, t_n, \hat{\mathbf{X}}_{t_n}^{(k)}, \hat{\mathbf{X}}_{t_N}(\hat{\mathbf{X}}_{t_n}^{(k)}, \theta)\Big), & n \in \{0, 1, \dots, \varrho-1\}, \\ \hat{\mathbf{X}}_{t_{n+1}}^{(k)} - \mathcal{H}\Big(t_{n+1}, t_n, \hat{\mathbf{X}}_{t_n}^{(k)}, \hat{\mathbf{X}}_{t_N}(\hat{\mathbf{X}}_{t_{\varrho-1}}^{(k)}, \theta)\Big), & n \in \{\varrho, \varrho+1, \dots, N-1\}. \end{cases} \tag{15}$$

Observe that this update rule applies a Jacobi-type iteration, where all components $\mathbf{X}_{t_1:t_N}^{(k+1)}$ are updated exclusively from the old components $\mathbf{X}_{t_1:t_N}^{(k)}$, resulting in delayed information propagation across time steps. Inspired by the classical Gauss-Seidel iteration, we propose a more efficient immediate update strategy wherein newly updated components are incorporated immediately into subsequent components' updating within the same iteration. Specifically, we first update the nonlinear-region components $\mathbf{X}_{t_n}^{(k)}$ for $n \in \{1, \cdots, \varrho\}$. Upon obtaining the refined nonlinear component $\mathbf{X}_{t_\varrho}^{(k+1)}$, we instantly propagate this refinement forward to compute the remaining linear-region components (i.e., $\mathbf{X}_{t_{\varrho+1}}^{(k+1)}, \dots, \mathbf{X}_{t_N}^{(k+1)}$). This approach ensures faster convergence by propagating the latest information forward, leading to the following improved update rule:

$$\hat{\mathbf{X}}_{t_0:t_N}^{(k+1)} = \hat{\mathbf{X}}_{t_0:t_N}^{(k)} - \mathcal{G}^{(k)}\mathcal{R}_{t_0:t_N}^{(k)}, \tag{16}$$

where the residual term $\mathcal{R}_{t_{n+1}}^{(k)}$ now incorporates the latest refined components:

$$\mathcal{R}_{t_{n+1}}^{(k)} = \begin{cases} \hat{\mathbf{X}}_{t_{n+1}}^{(k)} - \mathcal{H}\Big(t_{n+1}, t_n, \hat{\mathbf{X}}_{t_n}^{(k)}, \hat{\mathbf{X}}_{t_N}(\hat{\mathbf{X}}_{t_n}^{(k)}, \theta)\Big), n \in \{0, 1, \cdots, \varrho-1\}, \\ \hat{\mathbf{X}}_{t_{n+1}}^{(k)} - \mathcal{H}\Big(t_{n+1}, t_n, \hat{\mathbf{X}}_{t_n}^{(k+1)}, \hat{\mathbf{X}}_{t_N}(\hat{\mathbf{X}}_{t_{\varrho-1}}^{(k)}, \theta)\Big), n \in \{\varrho, \varrho+1, \cdots, N-1\}. \end{cases} \tag{17}$$

We observe that once the nonlinear components converge, the predicted clean sample $\hat{\mathbf{X}}_{t_N}(\hat{\mathbf{X}}_{t_{\varrho-1}}^{(k)}, \theta)$ becomes unchanged, implying that no further precise information can be propagated for the linear equations in subsequential iterations. To address this, we progressively convert the linear equations into nonlinear ones, such that the number of nonlinear equations remains unchanged as the iteration proceeds. Formally, at iteration $k$, suppose the unconverged components are $\hat{\mathbf{X}}_{t_n}^{(k)}, \cdots, \hat{\mathbf{X}}_{t_N}^{(k)}$, we maintain nonlinearity in the residuals $\{\mathcal{R}_{t_{i+1}}^{(k)}, i = n, n+1, \cdots, n+\varrho-1\}$ of the first $\varrho$ components through the following formulation:

$$\hat{\mathbf{X}}_{t_n:t_N}^{(k+1)} = \hat{\mathbf{X}}_{t_n:t_N}^{(k)} - \mathcal{G}^{(k)}\mathcal{R}_{t_n:t_N}^{(k)}, \tag{18}$$

where the residual term $\mathcal{R}_{t_{i+1}}^{(k)}$ can adaptively convert the linear equations into non-linear ones:

$$\mathcal{R}_{t_{i+1}}^{(k)} = \begin{cases} \hat{\mathbf{X}}_{t_{i+1}}^{(k)} - \mathcal{H}\Big(t_{i+1}, t_i, \hat{\mathbf{X}}_{t_i}^{(k)}, \hat{\mathbf{X}}_{t_N}(\hat{\mathbf{X}}_{t_i}^{(k)}, \theta)\Big), i \in \{n, n+1, \cdots, n+\varrho-1\}, \\ \hat{\mathbf{X}}_{t_{i+1}}^{(k)} - \mathcal{H}\Big(t_{i+1}, t_i, \hat{\mathbf{X}}_{t_i}^{(k+1)}, \hat{\mathbf{X}}_{t_N}(\hat{\mathbf{X}}_{t_{n+\varrho-1}}^{(k)}, \theta)\Big), i \in \{n+\varrho, n+\varrho+1, \cdots, N-1\}. \end{cases} \tag{19}$$

where $\mathcal{R}_{t_n}^{(k)} = \hat{\mathbf{X}}_{t_n}^{(k)} - \mathbf{X}_{t_n} = \mathbf{0}$. This approach ensures continuous refinement of $\hat{\mathbf{X}}_{t_N}(\hat{\mathbf{X}}_{t_{n+\varrho-1}}^{(k)}, \theta)$, yielding increasingly accurate solutions for the linear equations.

**Proposition 3** (Convergence Analysis). *The update rule in Eq. (18) achieves exact convergence to the generated sample from the sequential integral solver within $K \leqslant T$ iterations.*

*Proof.* See Appendix J for the complete derivation. $\square$

**Stopping Criterion.** Following existing works (Shih et al., 2024a; Tang et al., 2024; Lu et al., 2025), We define the stopping criterion as $\frac{1}{D}\Big\|\hat{\mathbf{X}}_{t_n}^{(k+1)} - \hat{\mathbf{X}}_{t_n}^{(k)}\Big\|_F^2 \leqslant \delta^2\sigma_{t_n}^2$, where $D$ is the dimensionality of the sample $\hat{\mathbf{X}}_{t_n}$, $\|\cdot\|_F$ denotes the Frobenius norm, and $\delta$ is an tolerance parameter.

---

**Algorithm 1:** ParaSolver-Turbo: a partial nonlinear parallel integral solver for diffusion models

---

**Input** : Diffusion model $S_\theta$, noise schedule $\alpha_t$ and $\sigma_t$, subinterval number $N$, preconditioning steps $M$, tolerance $\delta$, batch window size $p$, sample dimension $D$, nonlinear degree $\varrho$, maximum parallel step $K$.

**Output :** A sample.

1 Initialize $\{\hat{\mathbf{X}}_{t_n}^{(0)}, n = 0, \cdots, p\}$ by Eq. (20)    // Initialize with a few sampling steps.

2 $n, k \leftarrow 0, 0, k \in [0, K]$, and $n \in [0, N-1]$

3 **while** $n < N$ **do**

   4  $\quad \hat{\mathbf{X}}_{t_N}(\hat{\mathbf{X}}_{t_i}^{(k)}, \theta) = \frac{\hat{\mathbf{X}}_{t_i}^{(k)} - \sigma_{t_i} \mathbf{S}_\theta(\hat{\mathbf{X}}_{t_i}^{(k)}, t_i)}{\alpha_{t_i}}, \forall i \in \{n, \cdots, n + \varrho - 1\}.$ // Predict clean samples in parallel.

   5  $\quad \mathcal{R}_{t_{i+1}}^{(k)} = \hat{\mathbf{X}}_{t_{i+1}}^{(k)} - \mathcal{H}\left(t_{i+1}, t_i, \hat{\mathbf{X}}_{t_i}^{(k)}, \hat{\mathbf{X}}_{t_N}(\hat{\mathbf{X}}_{t_i}^{(k)}, \theta)\right), \forall i \in \{n, \cdots, n + \varrho - 1\}$ // Nonlinear residual.

   6  $\quad \hat{\mathbf{X}}_{t_{n+1}:t_{n+\varrho}}^{(k+1)} = \hat{\mathbf{X}}_{t_{n+1}:t_{n+\varrho}}^{(k)} - \mathcal{G}^{(k)} \mathcal{R}_{t_{n+1}:t_{n+\varrho}}^{(k)}$    // Update the nonlinear components.

   7  $\quad$ **for** $i \in \{n + \varrho, n + \varrho + 1, \cdots, n + p - 1\}$ **do**

   8  $\quad\quad \mathcal{R}_{t_{i+1}}^{(k)} = \hat{\mathbf{X}}_{t_{i+1}}^{(k)} - \mathcal{H}\left(t_{i+1}, t_i, \hat{\mathbf{X}}_{t_i}^{(k+1)}, \hat{\mathbf{X}}_{t_N}(\hat{\mathbf{X}}_{t_{n+\varrho-1}}^{(k)}, \theta)\right)$ in Eq. (17)   // Linear residual.

   9  $\quad\quad \hat{\mathbf{X}}_{t_{i+1}}^{(k+1)} = \hat{\mathbf{X}}_{t_{i+1}}^{(k)} - \mathcal{G}^{(k)} \mathcal{R}_{t_{i+1}}^{(k)}$   // Update the linear components using immediate updating strategy.

   10 $\quad$ Compute errors $\mathcal{E}_{t_{i+1}} = \frac{1}{D} \|\mathcal{R}_{t_{i+1}}^{(k)}\|_F^2, \forall i \in \{n, \cdots, n + p - 1\}$// Compute error for each component.

   11 $\quad s \leftarrow \min\limits_{i+1}\left(\{i + 1 | \mathcal{E}_{t_{i+1}} > \delta^2 \sigma_{t_{i+1}}^2, \forall i \in \{n, \cdots, n + p - 1\}\} \cup \{n + p\}\right)$    // The sliding stride.

   12 $\quad \hat{\mathbf{X}}_{t_{i+1}}^{(k+1)} = \mathcal{H}\left(t_{i+1}, t_i, \hat{\mathbf{X}}_{t_i}^{(k+1)}, \hat{\mathbf{X}}_{t_N}(\hat{\mathbf{X}}_{t_{n+\varrho-1}}^{(k)}, \theta)\right), \forall i \in \{n + p, \cdots, n + p + s - 1\}$   // Initialize new components for the next window.

   13 $\quad n \leftarrow n + s, \quad k \leftarrow k + 1, \quad p \leftarrow \min(p, N - n)$    // Update the iteration indexs.

**Return:** $\hat{\mathbf{X}}_{t_N}^{(K)}$

---

**Initialization.** To establish the diffusion trajectory, we adopt the ParaSolver methodology by performing $M$ steps of a sequential integral solver as preconditioning steps. The initialization process is formally expressed as:

$$\begin{cases} \hat{\mathbf{X}}_{t_n}^{(0)} = \mathbf{X}_{t_n}, & \text{if } 0 \leqslant n < M, \\ \hat{\mathbf{X}}_{t_n}^{(0)} = \mathcal{H}\left(t_n, t_{n-1}, \hat{\mathbf{X}}_{t_{n-1}}^{(0)}, \hat{\mathbf{X}}_{t_N}(\hat{\mathbf{X}}_{t_{M-1}}^{(0)}, \theta)\right), & \text{if } M \leqslant n < N, \end{cases} \tag{20}$$

where the initial state $\mathbf{X}_{t_0}$ is sampled from a standard Gaussian distribution, $\mathbf{X}_{t_0} \sim \mathcal{N}(\mathbf{0}, \mathbf{I})$. When $M = 0$, following Shih et al. (2024a), all samples are initialized with random noise.

**Sliding Window Strategy.** To optimize computational efficiency, ParaSolver utilizes a sliding window of fixed size $p$, retaining only the most recent $p$ denoised points $\{\hat{\mathbf{X}}_{t_n}^{(k)} : n = 0, \ldots, p - 1\}$ for parallel processing. This approach enhances parallel performance while adhering to GPU memory limitations. Adapted to our framework, the update rule becomes:

$$\hat{\mathbf{X}}_{t_n:t_{n+p}}^{(k+1)} = \hat{\mathbf{X}}_{t_n:t_{n+p}}^{(k)} - \mathcal{G}^{(k)} \mathcal{R}_{t_n:t_{n+p}}^{(k)}, \tag{21}$$

where the residual term $\mathcal{R}_{t_{i+1}}^{(k)}$ over a sliding window becomes:

$$\mathcal{R}_{t_{i+1}}^{(k)} = \begin{cases} \hat{\mathbf{X}}_{t_{i+1}}^{(k)} - \mathcal{H}\left(t_{i+1}, t_i, \hat{\mathbf{X}}_{t_i}^{(k)}, \hat{\mathbf{X}}_{t_N}(\hat{\mathbf{X}}_{t_i}^{(k)}, \theta)\right), & i \in \{n, \ldots, n + \varrho - 1\}, \\ \hat{\mathbf{X}}_{t_{i+1}}^{(k)} - \mathcal{H}\left(t_{i+1}, t_i, \hat{\mathbf{X}}_{t_i}^{(k+1)}, \hat{\mathbf{X}}_{t_N}(\hat{\mathbf{X}}_{t_{n+\varrho-1}}^{(k)}, \theta)\right), & i \in \{n + \varrho, \ldots, p - 1\}. \end{cases} \tag{22}$$

**Complete Algorithm.** Algorithm 1 outlines the full procedure of the proposed ParaSolver-Turbo. Initially, an array of starting guesses $\{\hat{\mathbf{X}}_{t_n}^{(0)} : n = 0, \ldots, p\}$ is prepared either through random noises or inexpensive preconditioning steps (Line 1). The parallel sampling loop then begins at Line 2. In Lines 4–6, all nonlinear equations are solved simultaneously. Subsequently, Lines 7–9 compute the solutions to the linear equations, leveraging the predicted clean samples obtained from the nonlinear solving phase. This step is computationally efficient since it avoids model evaluations. The discrepancy between the newly computed states and the current states is assessed at Line 10, followed by determining the feasible window sliding stride at Line 11. Line 12 initializes $s$ new points beyond the current window based on the computed stride.

Table 1: Comparisons on Stable Diffusion-v2, with classifier guidance w = 7.5. The visual comparisons are shown in *Appendix*.

| Steps | Method | Stable Diffusion-v2 | | | | | | |
|---|---|---|---|---|---|---|---|---|
| | | Iters ↓ | NFE ↓ | CLIP↑ | FID↓ | Time (s)↓ | | Speedup↑ |
| 1000 | DDPM | 1000 | 1000 | 25.6 | 55.9 | 128.0 | ■■ | 1.0× |
| | DDPM + ParaDiGMS | 65 | 2024 | 25.6 | 55.6 | 32.8 | ■ | 3.9× |
| | DDPM + ParaSolver | 32 | 1065 | 25.6 | 55.3 | 15.9 | ▮ | 8.1× |
| | DDPM + ParaSolver-Turbo | 32 | **195** | 25.6 | 55.7 | **3.2** | ▮ | **40.0×** |
| 50 | DDIM | 50 | 50 | 25.6 | 57.2 | 6.3 | ■ | 1.0× |
| | DDIM + ParaDiGMS | 21 | 132 | 25.6 | 56.9 | 3.3 | ■ | 1.9× |
| | DDIM + ParaSolver | 13 | 83 | 25.6 | 56.9 | 1.9 | ▮ | 3.3× |
| | DDIM + ParaSolver-Turbo | 13 | **73** | 25.6 | 57.1 | **1.6** | ▮ | **3.9×** |
| 25 | DDIM | 25 | 25 | 25.4 | 62.9 | 3.4 | ■ | 1.0× |
| | DDIM + ParaDiGMS | 18 | 53 | 25.4 | 62.8 | 3.2 | ■ | 1.2× |
| | DDIM + ParaSolver | 11 | 49 | 25.4 | 61.9 | 1.2 | ▮ | 2.8× |
| | DDIM + ParaSolver-Turbo | 11 | **42** | 25.4 | 61.8 | **1.1** | ▮ | **3.1×** |
| 50 | DPMSolver | 50 | 50 | 25.6 | 57.2 | 6.3 | ■ | 1.0× |
| | DPMSolver + ParaDiGMS | 25 | 132 | 25.6 | 57.2 | 3.3 | ■ | 1.8× |
| | DPMSolver + ParaSolver | 15 | 96 | 25.6 | 57.1 | 2.2 | ▮ | 3.1× |
| | DPMSolver + ParaSolver-Turbo | **14** | **93** | 25.6 | 57.1 | **1.7** | ▮ | **3.9×** |
| 25 | DPMSolver | 25 | 25 | 25.4 | 62.3 | 3.2 | ■ | 1.0× |
| | DPMSolver + ParaDiGMS | 15 | 81 | 25.4 | 62.3 | 2.2 | ■ | 1.5× |
| | DPMSolver + ParaSolver | 11 | 58 | 25.4 | 62.1 | 1.5 | ▮ | 2.1× |
| | DPMSolver + ParaSolver-Turbo | **10** | **55** | 25.4 | 62.1 | **1.2** | ▮ | **2.7×** |

## 6 EXPERIMENTS

**Evaluation Metrics.** We assessed ParaSolver-Turbo using five widely adopted metrics: function evaluation count (NFE), iterations (iters), wall-clock time, FID score (Heusel et al., 2017), and CLIP score (Hessel et al., 2021).

**Datasets and Models.** As per ParaSolver (Lu et al., 2025), we analyze our ParaSolver-Tubo across latent-space model (StableDiffusion-v2) and pixel-space model (LSUN Church) using the same settings.

**Algorithms**. We apply our approach to accelerate the performance of the state-of-the-art sequential sampling methods: DDPM (Ho et al., 2020), DDIM (Song et al., a), and DPMSolver (Lu et al., 2022). We evaluate the parallel efficiency of our ParaSolver-Turbo against ParaSolver and ParaDiGMS in accelerating the aforementioned three sequential methods as it incorporates all existing parallel methods.

**Hyperparameter Settings.** Following the previous settings in (Lu et al., 2025; Tang et al., 2024), we apply our ParaSolver-Turbo and ParaDiGMS to DDPM with 1000 sequential sampling steps. For DDIM and DPMSolver, we consider two settings: 25 and 50 sequential sampling steps. Besides, We follow the best tolerances for ParaDiGMS and ParaSolver in their papers. For our ParaSolver-Turbo implementation, we optimized the hyperparameters through grid search. On the StableDiffusion-v2 model, we set $\delta = 0.5$ for DDPM, while using $\delta = 0.03$ for both DDIM and DPMSolver. For the LSUN Church model, we use $\delta = 0.5$ for DDPM, and $\delta = 0.003$ for DDIM. Additionally, we set $\varrho = 8$ for ParaSolver-Turbo, as we utilize 8 NVIDIA 3090 GPUs. More details are provided in the *Appendix* D.

**Acceleration Comparison**. Table 1 shows the experimental results on DDPM, DDIM, DPMSolver, and their parallel variants when combined with ParaSolver, ParaDiGMS, and our ParaSolver-Turbo, respectively. ParaSolver-Turbo demonstrates significant acceleration for DDPM, DDIM, and DPM-Solver while maintaining comparable FID and CLIP scores. Notably, it achieves a remarkable wall-clock time speedup of up to $40.0\times$, setting a new benchmark for parallel diffusion methods. Although parallelization inherently trades computational efficiency for speed, leading to increased NFEs relative to sequential approaches, our method capitalizes on partial linearity to substantially

Table 2: The effect of the number of GPUs using LSUN Church. We report Batch Size for feeding into the model per GPU (BS), GPU Utilization (Util), and Speedup (Spd) for 1000 and 50 diffusion steps. We set $\varrho = 8$ for our method and $p = 24$ for all methods.

| Step | Method | 1 GPU | | | 4 GPUs | | | 8 GPUs | | |
|------|--------|----|------|------|----|------|------|----|------|------|
| | | BS | Util | Spd | BS | Util | Spd | BS | Util | Spd |
| 1000 | DDPM | 1 | 19% | 1.0× | 1 | 19% | 1.0× | 1 | 19% | 1.0× |
| | ParaDiGMS | 24 | 100% | 1.4× | 6 | 48% | 3.3× | 3 | 27% | 4.6× |
| | ParaSolver | 24 | 100% | 3.1× | 6 | 46% | 6.9× | 3 | 25% | 12.1× |
| | Ours | 8 | 59% | **8.3×** | 2 | 25% | **16.3×** | 1 | 19% | **43.7×** |
| 50 | DDPM | 1 | 19% | 1.0× | 1 | 19% | 1.0× | 1 | 19% | 1.0× |
| | ParaDiGMS | 24 | 98% | 0.4× | 6 | 41% | 0.7× | 3 | 23% | 1.6× |
| | ParaSolver | 24 | 98% | 0.8× | 6 | 40% | 1.1× | 3 | 24% | 2.9× |
| | Ours | 8 | 43% | **1.7×** | 2 | 24% | **2.6×** | 1 | 19% | **3.8×** |

reduce NFEs, thereby unleashing the parallel efficiency. Similarly, ParaSolver-Turbo consistently surpasses existing methods, delivering speed improvements of 2.7× to **50.0×** on LSUN Church model. The experimental results on the LSUN Church model are summarized in the *Appendix*.

**Generalization on VP, VE, and EDM Process.** We investigated the impact of the VP, VE, and EDM diffusion processes by converting the sequential sampler in Alg. 1 of work (Karras et al., 2022) into our parallel version. Adopting the Tab. 1 configurations specified in work (Karras et al., 2022), we then analyzed the effect of the nonlinear degree $\varrho$ on FID performance across these three processes using the provided ImageNet dataset. As shown in Tab. 3, our parallel sampling method demonstrates its applicability and effectiveness across all three diffusion processes. For each process, our parallel approach consistently reduces the number of sampling iterations while achieving comparable or improved FID scores compared to the sequential baseline. This highlights the robustness and efficiency of our method, making it a viable accelerator for various diffusion model formulations. We notice that our method yields a better FID score in the VP process. This improvement likely stems from the inherently stiff nature of the VP-ODE, combined with the step-parallel updates of our parallel solver. The step-parallel updates sever the temporal dependence chain across steps, thus effectively curtail the accumulation of truncation error across steps (Karras et al., 2022). As a result, our solver acts as an implicit numerical stabilizer for the stiff VP-ODE, which typically exhibits significant truncation error that degrades the performance of conventional VP samplers.

Table 3: The impact of nonlinear degree $\varrho$ on FID across EDM, VE, and VP processes over the ImageNet dataset. We set $\delta = 0.01$ for the EDM and VP processes, and $\delta = 0.1$ for the VE process.

| (a) EDM Process ($\delta = 0.01$) | | | (b) VE Process ($\delta = 0.1$) | | | (c) VP Process ($\delta = 0.01$) | | |
|--------|------|-------|--------|----------|-------|--------|---------|-------|
| **Method** | **FID** | **Iters** | **Method** | **FID** | **Iters** | **Method** | **FID** | **Iters** |
| Baseline | 11.1649 | 25 | Baseline | 423.0628 | 25 | Baseline | 104.15 | 25 |
| $\varrho = 4$ | 10.8811 | 22 | $\varrho = 4$ | 422.9794 | 14 | $\varrho = 4$ | 35.8963 | 12 |
| $\varrho = 6$ | 10.8718 | 21 | $\varrho = 6$ | 422.9293 | 14 | $\varrho = 6$ | 41.1196 | 10 |
| $\varrho = 8$ | 10.8621 | 20 | $\varrho = 8$ | 422.8247 | 13 | $\varrho = 8$ | 58.5588 | 8 |
| $\varrho = 10$ | 10.8626 | 15 | $\varrho = 10$ | 422.7672 | 10 | $\varrho = 10$ | 58.5215 | 7 |
| $\varrho = 12$ | 10.8618 | 12 | $\varrho = 12$ | 422.7651 | 9 | $\varrho = 12$ | 58.8340 | 7 |

**GPU Scaling Analysis.** Table 2 examines GPU scaling performance. Our method achieves 43.7× speedup, significantly outperforming existing approaches. Notably, our approach delivers outstanding performance even on a single GPU, achieving 8.3× acceleration. This single-GPU efficiency offers significant practical advantages, enabling high-speed generation for resource-constrained sce-

narios. The efficiency stems from our linearized sampling process that solves only a partial nonlinear system.

**Nonlinear Degree $\varrho$ Analysis**: Tab. 4 conducted an ablation study by accelerating a 50-step DPMSolver on the Stable Diffusion v2 model using 8 GPUs. As shown in the table, the FID and CLIP scores remain remarkably stable across a wide range of $\varrho$ values. This stability confirms our theoretical finding in Prop. 2 that $\varrho$ does not compromise the fidelity of the generated samples. Furthermore, the results reveal a clear trade-off between $\varrho$ and efficiency. While increasing $\varrho$ initially yields significant speedups, the gains diminish and eventually reverse. This is because a larger $\varrho$ requires more NFEs, as seen in the table. The increased NFE count results in a greater computational overhead for each GPU (proportional to NFEs / GPUs), which leads to a rapid increase in runtime after an optimal point (around $\varrho = 15$).

Table 4: Impact of varying $\varrho$ on generation quality (FID, CLIP) and performance (NFEs, Speedup).

| Method | FID | CLIP | Iters | NFEs | Speedup |
|---|---|---|---|---|---|
| Baseline | 57.23 | 25.64 | 50 | 50 | 1× |
| $\varrho = 5$ | 57.18 | 25.55 | 18 | 82 | 2.9× |
| $\varrho = 10$ | 57.25 | 25.59 | 12 | 101 | 3.0× |
| $\varrho = 15$ | 57.14 | 25.61 | 11 | 123 | 3.1× |
| $\varrho = 20$ | 57.22 | 25.60 | 12 | 160 | 2.6× |
| $\varrho = 25$ | 57.14 | 25.53 | 12 | 202 | 2.1× |
| $\varrho = 30$ | 57.06 | 25.58 | 12 | 223 | 2.1× |
| $\varrho = 35$ | 57.18 | 25.62 | 12 | 245 | 1.8× |
| $\varrho = 40$ | 57.15 | 25.61 | 12 | 264 | 1.8× |
| $\varrho = 45$ | 57.16 | 25.59 | 12 | 275 | 1.6× |
| $\varrho = 50$ | 57.14 | 25.57 | 12 | 284 | 1.6× |

## 7 CONCLUSION

In this work, we have presented ParaSolver-Turbo, a unified framework that reinterprets diffusion model sampling as a mixed system of linear and nonlinear equations. By exploiting the partial linear structure inherent in sequential sampling, ParaSolver-Turbo significantly reduces the computational overhead of prior parallel approaches while preserving sample quality. Theoretical analysis guarantees convergence to the sequential solver's trajectory with equal or fewer iterations, and extensive experiments demonstrate 2×–50× wall-clock time speedups without compromising fidelity.

## ETHICS & REPRODUCIBILITY STATEMENTS

Our work focuses on a fundamental algorithmic improvement for sampling from generative models and does not introduce new ethical concerns beyond those already associated with large-scale text-to-image models. The models used in our experiments are developed by third parties, and we use them as is. Our method could be used to accelerate the generation of harmful content, but it does not inherently make such generation easier or more likely than with standard samplers. We believe the primary positive impact is making high-fidelity generative AI more accessible for research and creative applications by lowering the inference time barrier. For reproducibility, we have detailed our methodology, algorithm, and experimental setup in the paper. We will release our source code, built upon standard open-source libraries, upon publication to allow for full verification of our results.

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

CONTENTS

## A  USE OF LLM

During the preparation of this work, we used Large Language Models (LLMs) to assist with the writing process. The primary uses included polishing and improving the fluency of the text, generating preliminary drafts of proofs, and assisting in the creation and formatting of tables. After using these tools, the author(s) reviewed and edited the content extensively. We take full responsibility for the entire content of this publication, including the ideas, proofs, and presentations ultimately contained in the final manuscript.

## B  LIMITATIONS, FUTURE DIRECTIONS, AND BROADER IMPACTS

Despite the promising results, our work has several limitations that open up avenues for future research.

- **Higher NFE than Sequential Methods:** Although our linearization method significantly reduces the Number of Function Evaluations (NFE) required for parallel sampling, the NFE

is still higher than that of sequential methods. Further reduction in NFE would substantially enhance parallel efficiency.

- **Computational Redundancy:** Compared to sequential sampling, our method requires more computational power. Considerable redundancy exists in the current parallel computation. Designing mechanisms, such as a cache, to minimize this redundant computation could further improve parallel efficiency.

- **Iterative Nature and Single-Step Generation:** Parallel sampling still remains an iterative correction process. A fascinating direction would be to investigate the distillation of parallel iterations or the development of a consistency model. This could potentially lead to few-step or even single-step parallel generation, representing a significant leap forward.

**The broader impacts.** The acceleration of diffusion models, while beneficial for creative applications and scientific research (e.g., molecular generation ), also carries the dual-use risks inherent in all powerful generative technologies. We acknowledge that faster model inference could potentially be misused for creating disinformation or other harmful content. We believe that by advancing the technical understanding of these models, our work can also contribute to the development of more robust detection and mitigation strategies.

## C  THE FRAMEWORK OF PARASOLVER-TURBO

Figure 1 shows the process of our ParaSolver-Turbo. The framework of ParaSolver-Turbo involves an iterative process for parallel noise prediction and state update. It starts by defining a window of size $p$ with $N$ time subintervals. Within this window, the algorithm identifies linear and nonlinear regions.

The process begins with an initial window of size $p$ and iteratively updates the state based on the latest samples. For each iteration $k$, the algorithm calculates residuals in parallel. These residuals are then used to update the state of the system.

The framework also includes an error check to determine convergence. If the error is within an acceptable range, the state is updated, and the window shifts to the next set of samples. If the error is not converged, the process continues with the next iteration. The window size is dynamically adjusted based on the remaining subintervals, ensuring efficient computation and accurate noise prediction.

## D  EXPERIMENTAL DETAILS

We evaluate ParaSolver-Turbo across various high-dimensional image generation models, such as the latent-space diffusion model StableDiffusion-v2 (Rombach et al., 2022) and the pixel-space diffusion model LSUN Church (Yu et al., 2015). The results of these experiments demonstrate that ParaSolver-Turbo enhances the efficiency of sequential sampling methods all while maintaining consistent sample quality as measured by metrics like FID score or CLIP score.

**Evaluation Metrics.** We assessed ParaSolver-Turbo using five widely adopted metrics: function evaluation count (NFE), iterations (iters), wall-clock time, FID score (Heusel et al., 2017), and CLIP score (Hessel et al., 2021).

**Datasets and Models.** As per ParaSolver (Lu et al., 2025), we analyze our ParaSolver-Tubo across latent-space model (StableDiffusion-v2) and pixel-space model (LSUN Church) using the same settings. For latent-space models, we leverage the StableDiffusion-v2 model. [2] (Russakovsky et al., 2015) . The StableDiffusion-v2 model generates images at a resolution of 768 by 768 pixels, utilizing a diffusion model that operates within a latent space dimension of 4 by 96 by 96. For pixel-space models, we employ models pretrained on the LSUN Church dataset [3] available on Huggingface (Ho et al., 2020; von Platen et al., 2022) . This LSUN Church pre-trained model operates in the pixel space with a resolution of 256 by 256. For all the computation of the FID score and CLIP score, we use the standard evaluation API [4] provided by Huggingface . For evaluation of the wall-clock

---

[2]https://huggingface.co/datasets/ILSVRC/imagenet-1k

[3]https://huggingface.co/google/ddpm-ema-church-256

[4]https://huggingface.co/docs/diffusers/conceptual/evaluation

time, we use the *torch.cuda.Event* [5] method provided by Pytorch 2.0 Paszke et al. (2019) . We assess the performance of all methods on 8 NVIDIA RTX 3090 GPUs, each equipped with 24268 MB of memory.

**Algorithms**. We apply our approach to accelerate the performance of the state-of-the-art sequential sampling methods: DDPM (Ho et al., 2020), DDIM (Song et al., a), and DPMSolver (Lu et al., 2022). We evaluate the parallel efficiency of our ParaSolver-Turbo against ParaSolver and ParaDiGMS in accelerating the aforementioned three sequential methods as it incorporates all existing parallel methods. ParaDiGMS[6] is implemented based on the the Diffusers library, and the other algorithms are all accessible in the widely-used library Diffusers (von Platen et al., 2022); hence we utilize them directly. However, we exclude the comparison with another parallel method (Tang et al., 2024) as it has not been integrated into the Diffusers library yet.

**Hyperparameter Settings.** Following the previous settings in (Lu et al., 2025; Tang et al., 2024), we apply our ParaSolver-Turbo and ParaDiGMS to DDPM with 1000 sequential sampling steps. For DDIM and DPMSolver, we consider two settings: 25 and 50 sequential sampling steps. as they are commonly utilized and capable of producing samples of similar quality to DDPM. Besides, We follow the best tolerances for ParaDiGMS and ParaSolver in their papers. For StableDiffusion-v2, ParaDiGMS for DDPM, DDIM, and DPMSolver need to set the tolerance as 0.5, 0.01, and 0.01 to achieve a similar sample quality as the corresponding sequential method, respectively; ParaSolver for DDPM, DDIM, and DPMSolver need to set the tolerance as 0.55, 0.01, and 0.01 to achieve a similar sample quality. For LSUN Church model, the tolerance of ParaDiGMS should be 0.5 and 0.001 for DDPM and DDIM; the tolerances of ParaSolver are set as 0.55 and 0.005 for DDPM and DDIM, respectively.

For our ParaSolver-Turbo implementation, we optimized the hyperparameters through grid search. On the StableDiffusion-v2 model, for DDPM, we search the best values on grid $\delta \in \{0.5, 0.6\}$; for DDIM and DPMSolver, we search the best values on grid $\delta \in \{0.05, 0.04, 0.03, 0.02, 0.01\}$. On the LSUN Church model, for DDPM, we search the best values on grid $\delta \in \{0.5, 0.6\}$; for DDIM and DPMSolver, we search the best values on grid $\delta \in \{0.005, 0.004, 0.003, 0.002, 0.001\}$.

# E    ACCELERATION FOR LSUN CHURCH MODEL

Similarly, ParaSolver-Turbo consistently surpasses existing methods, delivering speed improvements of 2.8× to 50.0× on the LSUN Church model across varying step configurations. At 1000 steps, ParaSolver-Turbo achieves a remarkable 50.0× speedup while maintaining identical FID (12.7), demonstrating its efficiency in high-fidelity generation. For 50-step and 25-step settings, it outperforms DDIM by 3.0× and 2.8×, respectively. Notably, ParaSolver-Turbo's parallelism significantly reduces iterations compared to baselines and minimizes NFE, underscoring its computational efficiency. This acceleration is critical for real-time applications, such as interactive design tools, where latency directly impacts user experience.

# F    VISION COMPARISION

This section shows the visual comparisons of when our ParaSolver-Turbo is applied to speed up DDPM, DDIM, and DPMSolver on Stable Diffusion v2. The results are shown in Figure 2, Figure 3, and Figure 4. We can see that our ParaSolver-Turbo significantly outperforms the competitors, with a faster speed to generate an image.

# G    VISION RESULTS

This section shows the visual results of when our ParaSolver-Turbo is applied to speed up DDPM, DDIM, and DPMSolver on Stable Diffusion v2. We generated 16 prompts with different styles by DeepSeek-v3 for each method. The results for DDIM plus ParaSolver-Turbo are shown in Figure 5. The results for DDPM plus ParaSolver-Turbo and the results for DPMSolver plus ParaSolver-Turbo are shown in supplementary.

---

[5]https://pytorch.org/docs/stable/generated/torch.cuda.Event.html
[6]https://github.com/AndyShih12/paradigms

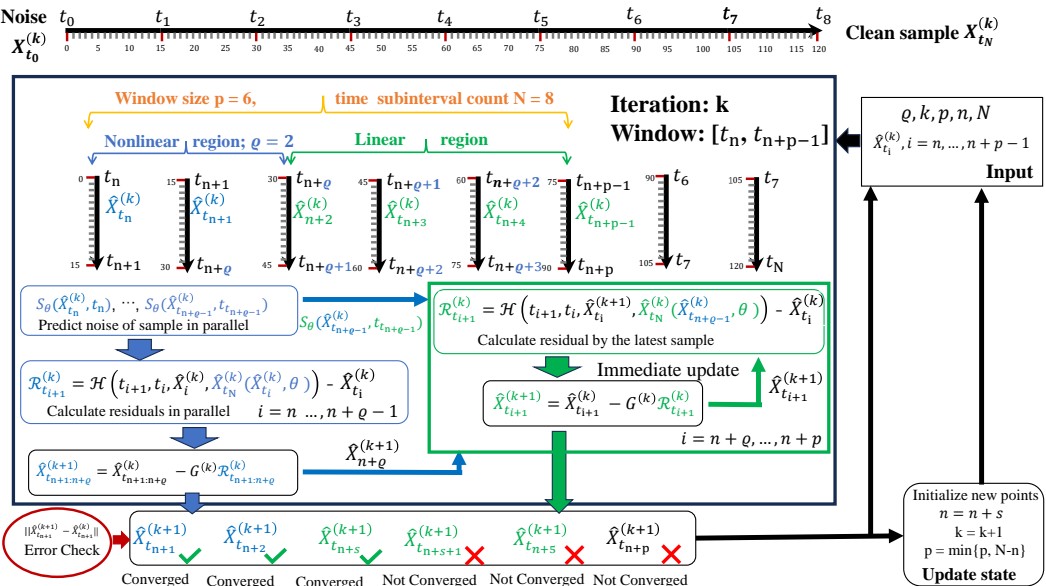

Figure 1: The Framework of ParaSolver-Turbo

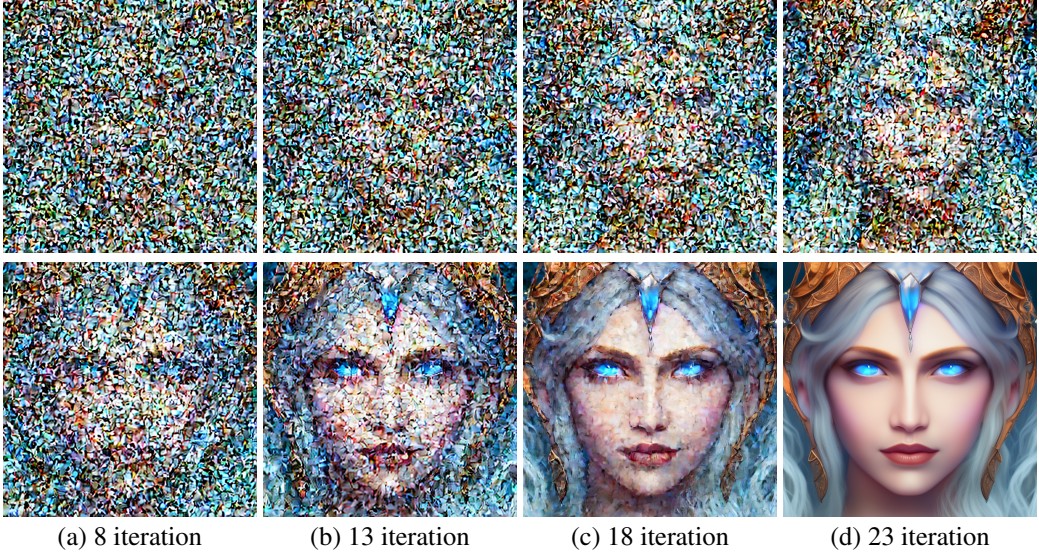

(a) 8 iteration  (b) 13 iteration  (c) 18 iteration  (d) 23 iteration

Figure 2: The intermediate generated images for expediting DDPM with 1000 sequential steps on Stable Diffusion Model v2. The images in the first row are produced by DDPM. The images in the second row are generated by our ParaSolver-Turbo. For our ParaSolver-Turbo, we set the number of subintervals $N$ as 1000, the preconditioning steps $M$ as 0, the tolerance $\delta$ as 0.5, the maximum parallel steps $K$ as 23, the parallel window size $p$ as 100, the nonlinear degree $\varrho$ as 8. The prompt is "*A stunning portrait of an ethereal elf queen with intricate silver jewelry, glowing blue eyes, flowing white hair, soft cinematic lighting, highly detailed, 8K.*".

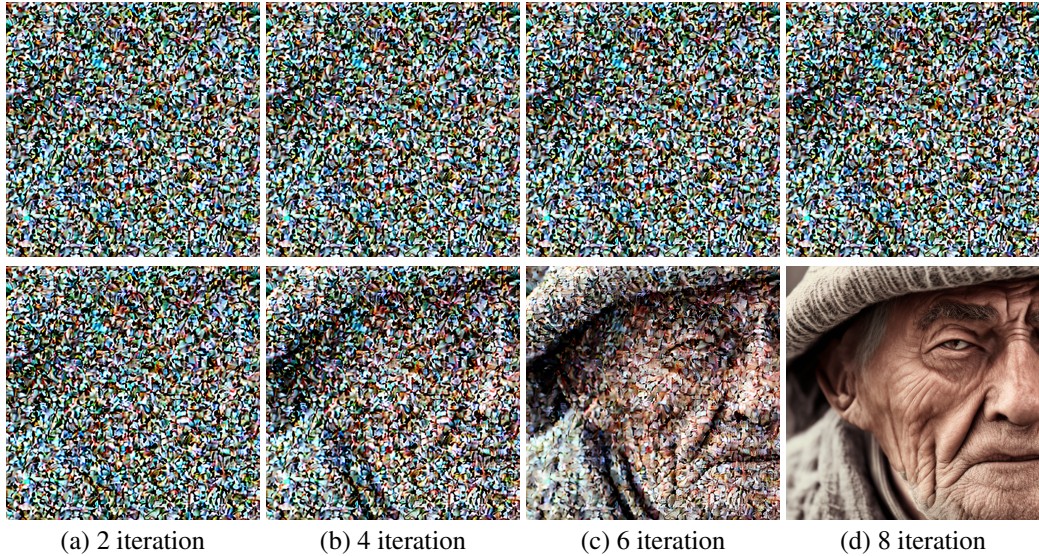

(a) 2 iteration      (b) 4 iteration      (c) 6 iteration      (d) 8 iteration

Figure 3: The intermediate generated images for expediting DDIM with 50 sequential steps on Stable Diffusion Model v2. The images in the first row are produced by DDIM. The images in the second row are generated by ParaSolver-Turbo. For our ParaSolver-Turbo, we set the number of subintervals $N$ as 50, the preconditioning steps $M$ as 0, the tolerance $\delta$ as 0.03, the maximum parallel steps $K$ as 11, the parallel window size $p$ as 50, the nonlinear degree $\varrho$ as 8. The prompt is "*A highly detailed portrait of an elderly man with wrinkles, wearing a traditional woolen hat, cinematic lighting, 8K, ultra-realistic, photorealistic, depth of field, soft shadows, film grain*" .

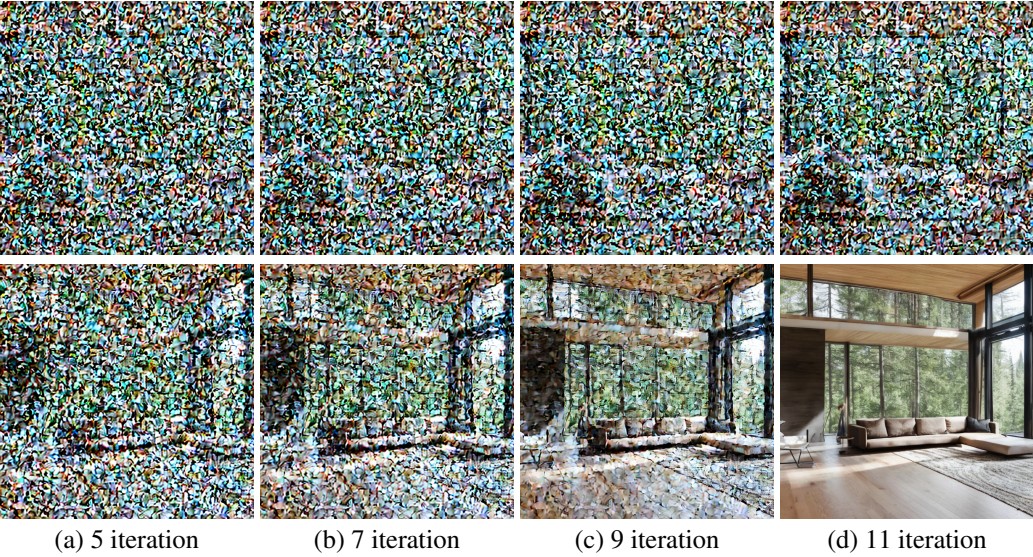

(a) 5 iteration      (b) 7 iteration      (c) 9 iteration      (d) 11 iteration

Figure 4: The intermediate generated images for expediting DPMSolver with 50 sequential steps on Stable Diffusion Model v2. The images in the first row are produced by DPMSolver. The images in the second row are generated by ParaSolver-Turbo. For our ParaSolver-Turbo, we set the number of subintervals $N$ as 50, the preconditioning steps $M$ as 0, the tolerance $\delta$ as 0.03, the maximum parallel steps $K$ as 11, the parallel window size $p$ as 50, the nonlinear degree $\varrho$ as 8. The prompt is "*A modern minimalist living room with floor-to-ceiling windows overlooking a forest, Scandinavian design, natural wood and neutral tones, soft daylight.*" .

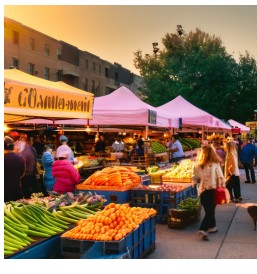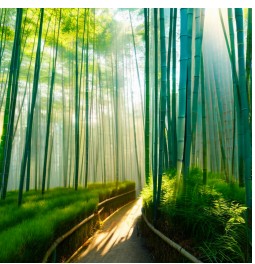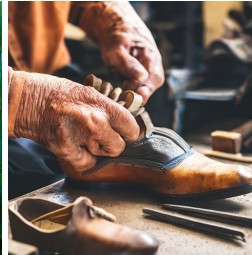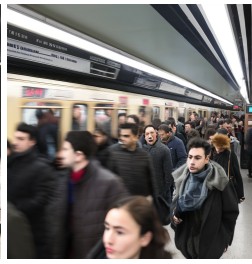

(a) A bustling farmer's market at sunset with vibrant fruits and vegetables on display, people chatting, and warm golden light

(b) A misty morning in a Japanese bamboo forest with sunlight filtering through the tall green stalks

(c) An elderly craftsman hand-making leather shoes in a small workshop filled with tools and natural light

(d) A crowded subway station during rush hour with commuters wearing winter coats and carrying briefcases

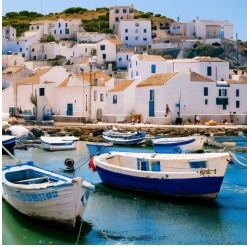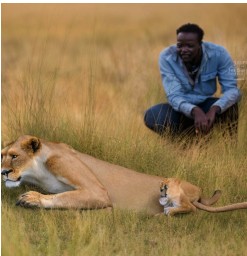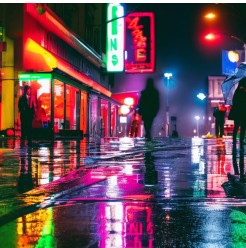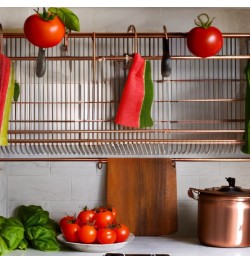

(e) A Mediterranean coastal village with whitewashed houses, blue shutters, and fishing boats in the harbor

(f) A wildlife photographer crouching in tall grass to capture a lioness with her cubs on the African savanna

(g) A rainy city street at night with colorful neon signs reflecting on wet pavement and people with umbrellas

(h) A traditional Italian kitchen with fresh pasta drying on racks, tomatoes, basil, and copper pots hanging

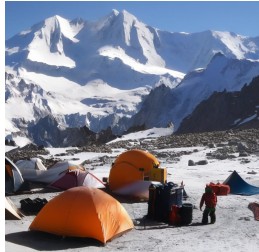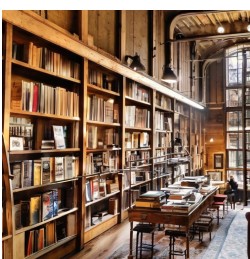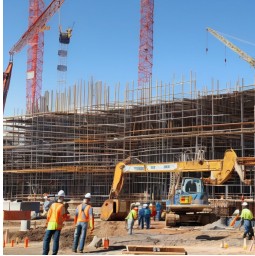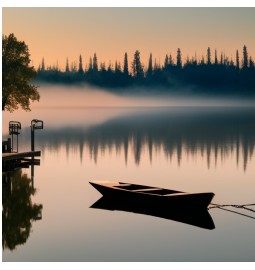

(i) A high-altitude mountain base camp with tents, climbers in heavy gear, and snow-capped peaks in background

(j) A vintage bookstore with floor-to-ceiling wooden shelves, ladder, and sunlight streaming through windows

(k) A busy construction site with workers in hard hats, cranes, and half-built steel structure against blue sky

(l) A peaceful lakeside dock at dawn with mist rising off still water and a single wooden rowboat tied up

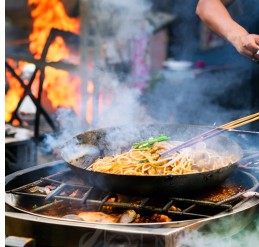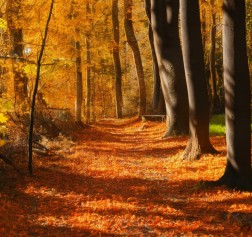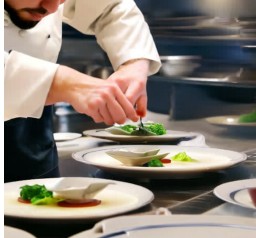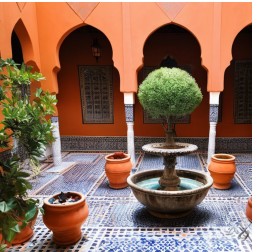

(m) A street food vendor in Bangkok cooking pad thai on a sizzling wok with smoke and aromatic steam rising

(n) An autumn forest path covered in fallen orange leaves with sunlight creating dappled patterns through trees

(o) A professional chef plating an elegant dish in a high-end restaurant kitchen with precise attention to detail

(p) A traditional Moroccan riad courtyard with intricate tile work, fountain, and potted orange trees

Table 5: Quantitative comparisons of different methods on LSUN Church over 5000 samples. DPM-Solver is not integrated with the LSUN model in the Diffusers library; thus we exclude it.

| Steps | Method | LSUN Church | | | | |
|---|---|---|---|---|---|---|
| | | Iters ↓ | NFE ↓ | FID↓ | Time (s)↓ | Speedup↑ |
| 1000 | DDPM | 1000 | 1000 | 12.7 | 50.0 | 1.0× |
| | DDPM + ParaDiGMS | 65 | 2082 | 12.7 | 10.8 | 4.6× |
| | DDPM + ParaSolver | 42 | 1079 | 12.7 | 4.1 | 12.1× |
| | DDPM + ParaSolver-Turbo | 42 | 196 | 12.7 | **1.0** | **50.0×** |
| 50 | DDIM | 50 | 50 | 15.5 | 1.8 | 1.0× |
| | DDIM + ParaDiGMS | 23 | 202 | 15.8 | 1.5 | 1.2× |
| | DDIM + ParaSolver | 14 | 73 | 15.7 | 0.8 | 2.2× |
| | DDIM + ParaSolver-Turbo | 14 | 67 | 15.7 | **0.6** | **3.0×** |
| 25 | DDIM | 25 | 25 | 15.6 | 1.1 | 1.0× |
| | DDIM + ParaDiGMS | 15 | 96 | 15.7 | 0.8 | 1.4× |
| | DDIM + ParaSolver | 10 | 44 | 15.7 | 0.5 | 2.2× |
| | DDIM + ParaSolver-Turbo | 10 | 39 | 15.7 | **0.4** | **2.8×** |

## H    PROOF OF PROPOSITION 1

Before proceeding with the proof, we first state the following practical assumption regarding the denoising model.

**Assumption 1.** *The denoising network $\mathbf{S}_{\theta*}(\mathbf{X}_t, t)$ is a well-trained noise predictor. Specifically, for any noisy sample $\mathbf{X}_t = \alpha_t \mathbf{X}_T + \sigma_t \epsilon_t$ (where $\mathbf{X}_T$ is the clean sample and $\epsilon_t \sim \mathcal{N}(0, \mathbf{I})$ is the specific noise instance), there exists a upper bound $\zeta > 0$ such that the model's prediction error is bounded:*

$$\|\mathbf{S}_{\theta*}(\mathbf{X}_t, t) - \epsilon_t\| < \zeta$$

We recall the forward process $p(\mathbf{X}_t|\mathbf{X}_T) = \mathcal{N}(\alpha_t \mathbf{X}_T, \sigma_t^2 \mathbf{I})$, where $\alpha_t$ and $\sigma_t$ are the noise schedule coefficients. We define the log-SNR as $\lambda_t = \log(\alpha_t/\sigma_t)$. It follows that $e^{-\lambda_t} = \sigma_t/\alpha_t$ and $\lambda_t$ is a strictly decreasing function of $t$.

Following Proposition 3.1 in DPM-Solver (Lu et al., 2022), the exact solution of the corresponding SDE from time $s$ to $t$ is given by:

$$\Phi(t, s, \mathbf{X}_s) = \frac{\alpha_t}{\alpha_s}\mathbf{X}_s + \alpha_t \int_{\lambda_s}^{\lambda_t} e^{-\lambda}\sigma_\lambda \nabla_{\mathbf{X}_\lambda} \log p(\mathbf{X}_\lambda)d\lambda$$
$$+ \int_{\lambda_s}^{\lambda_t} g(\lambda)d\mathbf{W}, \tag{23}$$

where $p(\mathbf{X}_\lambda)$ is the marginal distribution. In a practical solver, the true score $\nabla_{\mathbf{X}_\lambda} \log p(\mathbf{X}_\lambda) = -\mathbb{E}[\epsilon|\mathbf{X}_\lambda]/\sigma_\lambda$ is unavailable and is replaced by the model's prediction. Specifically, the solver approximates the score using the noise prediction model $\mathbf{S}_{\theta*}$:

$$\hat{\nabla}_{\mathbf{X}_\lambda} \log p(\mathbf{X}_\lambda) = -\frac{\mathbf{S}_{\theta*}(\mathbf{X}_\lambda, \lambda)}{\sigma_\lambda}$$

By substituting this approximation into Eq. (23), we define the **practical sequential solver** $\hat{\Phi}$ as:

$$\hat{\Phi}(t, s, \mathbf{X}_s) = \frac{\alpha_t}{\alpha_s}\mathbf{X}_s + \alpha_t \int_{\lambda_s}^{\lambda_t} e^{-\lambda}\sigma_\lambda \left(-\frac{\mathbf{S}_{\theta*}(\mathbf{X}_\lambda, \lambda)}{\sigma_\lambda}\right) d\lambda$$
$$+ \int_{\lambda_s}^{\lambda_t} g(\lambda)d\mathbf{W} \tag{24}$$
$$= \frac{\alpha_t}{\alpha_s}\mathbf{X}_s - \alpha_t \int_{\lambda_s}^{\lambda_t} e^{-\lambda}\mathbf{S}_{\theta*}(\mathbf{X}_\lambda, \lambda)d\lambda + \int_{\lambda_s}^{\lambda_t} g(\lambda)d\mathbf{W}.$$

Our goal is to show that this practical solver $\hat{\Phi}$ can be precisely approximated by a linear solver. We use Assumption 1 to decompose the model's prediction $\mathbf{S}_{\theta*}$ into the ground-truth noise $\epsilon_\lambda$ and a bounded error term $\delta(\mathbf{X}_\lambda, \lambda)$:

$$\mathbf{S}_{\theta*}(\mathbf{X}_\lambda, \lambda) = \epsilon_\lambda + \delta(\mathbf{X}_\lambda, \lambda),$$

where $\|\delta(\mathbf{X}_\lambda, \lambda)\| < \zeta$.

Substituting this decomposition back into our practical solver Eq. (24):

$$\hat{\Phi}(t, s, \mathbf{X}_s) = \frac{\alpha_t}{\alpha_s}\mathbf{X}_s - \alpha_t \int_{\lambda_s}^{\lambda_t} e^{-\lambda}\left(\epsilon_\lambda + \delta(\mathbf{X}_\lambda, \lambda)\right) d\lambda$$
$$+ \int_{\lambda_s}^{\lambda_t} g(\lambda)d\mathbf{W}$$
$$= \underbrace{\frac{\alpha_t}{\alpha_s}\mathbf{X}_s - \alpha_t \int_{\lambda_s}^{\lambda_t} e^{-\lambda}\epsilon_\lambda d\lambda + \int_{\lambda_s}^{\lambda_t} g(\lambda)d\mathbf{W}}_{\mathcal{H}(t, s, \mathbf{X}_s, \mathbf{X}_T)} \tag{25}$$
$$\underbrace{- \alpha_t \int_{\lambda_s}^{\lambda_t} e^{-\lambda}\delta(\mathbf{X}_\lambda, \lambda)d\lambda}_{\mathcal{Z}}.$$

We can identify the first part, $\mathcal{H}$, as the ideal linear solver. By substituting the definition of the ground-truth noise, $\epsilon_\lambda = (\mathbf{X}_\lambda - \alpha_\lambda \mathbf{X}_T(\mathbf{X}_\lambda))/\sigma_\lambda$, this term becomes:

$$\mathcal{H}(t, s, \mathbf{X}_s, \mathbf{X}_T) = \frac{\alpha_t}{\alpha_s}\mathbf{X}_s - \alpha_t \int_{\lambda_s}^{\lambda_t} e^{-\lambda}\frac{\mathbf{X}_\lambda - \alpha_\lambda \mathbf{X}_T(\mathbf{X}_\lambda)}{\sigma_\lambda}d\lambda$$
$$+ \int_{\lambda_s}^{\lambda_t} g(\lambda)d\mathbf{W}, \tag{26}$$

which is the linear solver structure presented in the proposition.

The second part, $\mathcal{Z}$, represents the cumulative error introduced by the non-ideal denoising model:

$$\mathcal{Z} = -\alpha_t \int_{\lambda_s}^{\lambda_t} e^{-\lambda}\delta(\mathbf{X}_\lambda, \lambda)d\lambda$$

We can bound the norm of this error term using the triangle inequality for integrals and Assumption 1:

$$\|\mathcal{Z}\| = \left\|-\alpha_t \int_{\lambda_s}^{\lambda_t} e^{-\lambda}\delta(\mathbf{X}_\lambda, \lambda)d\lambda\right\|$$
$$\leqslant \alpha_t \int_{\lambda_s}^{\lambda_t} e^{-\lambda}\|\delta(\mathbf{X}_\lambda, \lambda)\|d\lambda$$
$$\leqslant \alpha_t\zeta \int_{\lambda_s}^{\lambda_t} e^{-\lambda}d\lambda \tag{27}$$
$$= \alpha_t\zeta \left[-e^{-\lambda}\right]_{\lambda_s}^{\lambda_t}$$
$$= \alpha_t\zeta \left(-e^{-\lambda_t} - (-e^{-\lambda_s})\right)$$
$$= \alpha_t\zeta \left(e^{-\lambda_s} - e^{-\lambda_t}\right).$$

By substituting the definition $e^{-\lambda} = \sigma/\alpha$, we get the final bound:

$$\|\mathcal{Z}\| \leqslant \alpha_t\zeta \left(\frac{\sigma_s}{\alpha_s} - \frac{\sigma_t}{\alpha_t}\right) = \zeta\sigma_t \left(\frac{\alpha_t\sigma_s}{\alpha_s\sigma_t} - 1\right)$$

Given that $s$ and $t$ represent consecutive time steps (i.e., the step size $s - t$ is small), we have $t \approx s$, $\alpha_t \approx \alpha_s$, and $\sigma_t \approx \sigma_s$. As the step size approaches zero, the bound approaches:

$$\lim_{t \to s} \int_{t_0}^{t_1} \zeta\sigma_t \left(\frac{\alpha_t\sigma_s}{\alpha_s\sigma_t} - 1\right) dt = \zeta(t_1 - t_0).$$

Since $\zeta$ is assumed to be small (for a well-trained model); $t_1 - t_0$ is also a limited value denoting the integral time range; and the term $\zeta\sigma_t \left(\frac{\alpha_t\sigma_s}{\alpha_s\sigma_t} - 1\right)$ is small for typical step sizes, the error $\|\mathcal{Z}\|$ is negligible. This demonstrates that the practical sequential solver $\hat{\Phi}$ is well-approximated by the linear solver $\mathcal{H}$, completing the proof.

## I    PROOF OF PROPOSITION 2

*Proof.* Let $\{\mathbf{X}_{t_n}\}_{n=0}^N$ be the exact trajectory generated by the sequential integral solver. From Proposition 1, the step transition is given by:

$$\mathbf{X}_{t_{n+1}} = \mathcal{H}(t_{n+1}, t_n, \mathbf{X}_{t_n}, \mathbf{X}_T) + \mathcal{Z}_n, \tag{28}$$

where $\mathbf{X}_T$ is the ground-truth data and $\|\mathcal{Z}_n\| \approx 0$ is the linearization error. Let $\{\hat{\mathbf{X}}_{t_n}\}_{n=0}^N$ be the solution to the $\varrho$-nonlinear system with the update rule:

$$\hat{\mathbf{X}}_{t_{n+1}} = \mathcal{H}(t_{n+1}, t_n, \hat{\mathbf{X}}_{t_n}, \mathbf{E}_{t_n}). \tag{29}$$

Here, $\mathbf{E}_{t_n}$ denotes the estimated clean sample. Based on Definition 2, this estimator is explicitly defined as:

$$\mathbf{E}_{t_n} = \begin{cases} \hat{\mathbf{X}}_{t_N}(\hat{\mathbf{X}}_{t_n}, \theta) & \text{if } n < \varrho, \\ \hat{\mathbf{X}}_{t_N}(\hat{\mathbf{X}}_{t_{\varrho-1}}, \theta) & \text{if } n \geqslant \varrho. \end{cases} \tag{30}$$

We define the cumulative state error as $\boldsymbol{\Delta}_{t_n} = \hat{\mathbf{X}}_{t_n} - \mathbf{X}_{t_n}$ and the clean sample estimation error as $\boldsymbol{\xi}_n = \mathbf{E}_{t_n} - \mathbf{X}_T$. To derive the error recurrence, we subtract Eq. (28) from Eq. (29). Using the integral definition of $\mathcal{H}$ from Eq. (7), we obtain:

$$\boldsymbol{\Delta}_{t_{n+1}} = \frac{\alpha_{t_{n+1}}}{\alpha_{t_n}}\boldsymbol{\Delta}_{t_n} - \alpha_{t_{n+1}}\int_{\lambda_{t_n}}^{\lambda_{t_{n+1}}} e^{-\lambda}\left(\frac{\hat{\mathbf{X}}_\lambda - \alpha_\lambda \mathbf{E}_{t_n}}{\sigma_\lambda} - \frac{\mathbf{X}_\lambda - \alpha_\lambda \mathbf{X}_T}{\sigma_\lambda}\right)d\lambda - \mathcal{Z}_n. \quad (31)$$

Using the Log-SNR property that $e^{-\lambda} = \sigma_\lambda/\alpha_\lambda$, the integrand simplifies as follows:

$$\frac{\sigma_\lambda}{\alpha_\lambda}\left(\frac{\hat{\mathbf{X}}_\lambda - \mathbf{X}_\lambda}{\sigma_\lambda} - \frac{\alpha_\lambda(\mathbf{E}_{t_n} - \mathbf{X}_T)}{\sigma_\lambda}\right) = \frac{\boldsymbol{\Delta}_\lambda}{\alpha_\lambda} - \boldsymbol{\xi}_n. \quad (32)$$

Substituting this back into the error equation yields:

$$\boldsymbol{\Delta}_{t_{n+1}} = \frac{\alpha_{t_{n+1}}}{\alpha_{t_n}}\boldsymbol{\Delta}_{t_n} - \alpha_{t_{n+1}}\int_{\lambda_{t_n}}^{\lambda_{t_{n+1}}}\left(\frac{\boldsymbol{\Delta}_\lambda}{\alpha_\lambda} - \boldsymbol{\xi}_n\right)d\lambda - \mathcal{Z}_n. \quad (33)$$

To evaluate the intermediate error $\boldsymbol{\Delta}_\lambda$ inside the integral, we invoke the **Superposition Principle** for linear differential equations (Boyce et al., 2021; Chen, 1984). The total error $\boldsymbol{\Delta}_\lambda$ can be rigorously decomposed into the sum of the *Zero-Input Response* (Homogeneous) and the *Zero-State Response* (Forced):

- **Homogeneous Response**: This component represents the natural evolution of the initial state error $\boldsymbol{\Delta}_{t_n}$ through the system dynamics, independent of the external estimation error. Due to the signal scaling inherent in the diffusion process $\mathcal{H}$, this is given by:

$$\boldsymbol{\Delta}_\lambda^{\text{homo}} = \frac{\alpha_\lambda}{\alpha_{t_n}}\boldsymbol{\Delta}_{t_n}. \quad (34)$$

- **Forced Response**: This component captures the error accumulated over the interval $[\lambda_{t_n}, \lambda]$ solely due to the input source $\boldsymbol{\xi}_n$. Based on the integral structure of the linear solver in Eq. (7), the contribution accumulates as:

$$\boldsymbol{\Delta}_\lambda^{\text{force}} = \alpha_\lambda\left[\int_{\lambda_{t_n}}^{\lambda} e^{-\tau}\frac{\alpha_\tau}{\sigma_\tau}d\tau\right]\boldsymbol{\xi}_n. \quad (35)$$

Substituting $e^{-\tau} = \sigma_\tau/\alpha_\tau$, the integrand simplifies perfectly to unity:

$$\boldsymbol{\Delta}_\lambda^{\text{force}} = \alpha_\lambda\left[\int_{\lambda_{t_n}}^{\lambda} 1 d\tau\right]\boldsymbol{\xi}_n = \alpha_\lambda(\lambda - \lambda_{t_n})\boldsymbol{\xi}_n. \quad (36)$$

Combining these components via superposition, we obtain the explicit expression for the intermediate error:

$$\boldsymbol{\Delta}_\lambda = \underbrace{\frac{\alpha_\lambda}{\alpha_{t_n}}\boldsymbol{\Delta}_{t_n}}_{\text{Homogeneous Response}} + \underbrace{\alpha_\lambda(\lambda - \lambda_{t_n})\boldsymbol{\xi}_n}_{\text{Forced Response}}. \quad (37)$$

We substitute this expression for $\boldsymbol{\Delta}_\lambda$ into the term $\frac{\boldsymbol{\Delta}_\lambda}{\alpha_\lambda}$. This leads to an algebraic cancellation of the time-dependent coefficient $\alpha_\lambda$ in the homogeneous term:

$$\frac{\boldsymbol{\Delta}_\lambda}{\alpha_\lambda} = \frac{1}{\alpha_\lambda}\left(\frac{\alpha_\lambda}{\alpha_{t_n}}\boldsymbol{\Delta}_{t_n} + \alpha_\lambda(\lambda - \lambda_{t_n})\boldsymbol{\xi}_n\right) = \frac{1}{\alpha_{t_n}}\boldsymbol{\Delta}_{t_n} + (\lambda - \lambda_{t_n})\boldsymbol{\xi}_n. \quad (38)$$

Now we can analytically evaluate the integral in Eq. (33):

$$\int_{\lambda_{t_n}}^{\lambda_{t_{n+1}}}\left(\frac{\boldsymbol{\Delta}_\lambda}{\alpha_\lambda} - \boldsymbol{\xi}_n\right)d\lambda = \int_{\lambda_{t_n}}^{\lambda_{t_{n+1}}}\left(\frac{\boldsymbol{\Delta}_{t_n}}{\alpha_{t_n}} + (\lambda - \lambda_{t_n})\boldsymbol{\xi}_n - \boldsymbol{\xi}_n\right)d\lambda$$

$$= \frac{\Delta\lambda_n}{\alpha_{t_n}}\boldsymbol{\Delta}_{t_n} + \left[\frac{(\lambda - \lambda_{t_n})^2}{2} - (\lambda - \lambda_{t_n})\right]_{\lambda_{t_n}}^{\lambda_{t_{n+1}}}\boldsymbol{\xi}_n \quad (39)$$

$$= \frac{\Delta\lambda_n}{\alpha_{t_n}}\boldsymbol{\Delta}_{t_n} + \left(\frac{(\Delta\lambda_n)^2}{2} - \Delta\lambda_n\right)\boldsymbol{\xi}_n,$$

where $\Delta\lambda_n = \lambda_{t_{n+1}} - \lambda_{t_n}$.

Plugging this integral result back into the expression for $\boldsymbol{\Delta}_{t_{n+1}}$ and grouping the terms by error source:

$$\boldsymbol{\Delta}_{t_{n+1}} = \underbrace{\frac{\alpha_{t_{n+1}}}{\alpha_{t_n}}\left(1 - \Delta\lambda_n\right)\boldsymbol{\Delta}_{t_n}}_{\mathcal{A}_n} + \underbrace{\alpha_{t_{n+1}}\Delta\lambda_n\left(1 - \frac{\Delta\lambda_n}{2}\right)\boldsymbol{\xi}_n}_{\mathcal{B}_n} - \mathscr{Z}_n. \tag{40}$$

**Analysis of Coefficients**: We analyze the magnitude of the coefficients $\mathcal{A}_n$ and $\mathcal{B}_n$ to establish convergence.

First, for the stability coefficient $\mathcal{A}_n$, we utilize the identity $\Delta\lambda_n = \log\frac{\alpha_{t_{n+1}}\sigma_{t_n}}{\alpha_{t_n}\sigma_{t_{n+1}}}$ (since we define the log-SNR as $\lambda_t = \log(\alpha_t/\sigma_t)$). The term $(1 - \Delta\lambda_n)$ can be expressed as a logarithmic difference:

$$1 - \Delta\lambda_n \equiv \log\frac{\alpha_{t_{n+1}}\sigma_{t_n}}{\alpha_{t_{n+1}}\sigma_{t_n}} - \log\frac{\alpha_{t_{n+1}}\sigma_{t_n}}{\alpha_{t_n}\sigma_{t_{n+1}}} = \log\frac{\alpha_{t_n}\sigma_{t_{n+1}}}{\alpha_{t_{n+1}}\sigma_{t_n}}. \tag{41}$$

Since the time step size is generally small, the ratio inside the logarithm is close to 1, making this term approach 0. Thus, $\mathcal{A}_n \approx 0$, confirming that the historical error is rapidly dampened.

Second, for the estimation error coefficient $\mathcal{B}_n$, we apply a similar transformation to the term $(1 - \frac{1}{2}\Delta\lambda_n)$:

$$1 - \frac{1}{2}\Delta\lambda_n \equiv \log\frac{\alpha_{t_{n+1}}\sigma_{t_n}}{\alpha_{t_{n+1}}\sigma_{t_n}} - \log\left(\frac{\alpha_{t_{n+1}}\sigma_{t_n}}{\alpha_{t_n}\sigma_{t_{n+1}}}\right)^{1/2} = \log\left(\sqrt{\frac{\alpha_{t_n}\sigma_{t_{n+1}}}{\alpha_{t_{n+1}}\sigma_{t_n}}}\right). \tag{42}$$

In a similar vein, this logarithmic term approaches 0. Consequently, the entire coefficient $\mathcal{B}_n$ is the product of bounded terms and this vanishing log-difference:

$$\mathcal{B}_n = \alpha_{t_{n+1}}\Delta\lambda_n\left(1 - \frac{\Delta\lambda_n}{2}\right) \approx 0. \tag{43}$$

**Conclusion**: The error recurrence bound is established as:

$$\|\boldsymbol{\Delta}_{t_{n+1}}\| \leqslant |\mathcal{A}_n|\|\boldsymbol{\Delta}_{t_n}\| + |\mathcal{B}_n|\|\boldsymbol{\xi}_n\| + \|\mathscr{Z}_n\|. \tag{44}$$

Regardless of the value of $\varrho$ (which determines the source of $\mathbf{E}_{t_n}$), the non-ideal estimator $\mathbf{E}_{t_n}$ introduces an error $\boldsymbol{\xi}_n$. However, this estimation error is effectively suppressed by the coefficient $\mathcal{B}_n$, which approaches zero. Given that the initial error $\|\boldsymbol{\Delta}_{t_0}\| = 0$, and the terms $|\mathcal{A}_n|$, $|\mathcal{B}_n|$, and $\|\mathscr{Z}_n\|$ are all negligible, the total trajectory deviation $\boldsymbol{\Delta}_{t_{n+1}}$ remains negligible throughout the sampling process. Thus, all $\varrho$-nonlinear systems are statistically equivalent. $\qquad\square$

## J   PROOF OF CONVERGENCE IN PROPOSITION 3

Assume by induction that for iteration $k$, the first $n$ components $\{\hat{\mathbf{X}}_{t_i}^{(k)}\}_{i=0}^{n-1}$ have converged to the sequential solution $\{\mathbf{X}_{t_i}\}_{i=0}^{n-1}$ from Proposition 1. Then for the next component $\hat{\mathbf{X}}_{t_n}^{(k+1)}$, we have:

$$\begin{aligned}
\hat{\mathbf{X}}_{t_n}^{(k+1)} &= \hat{\mathbf{X}}_{t_n}^{(k)} - \mathcal{G}^{(k)}\mathcal{R}_{t_n}^{(k)} \\
&= \hat{\mathbf{X}}_{t_n}^{(k)} - \mathcal{G}^{(k)}\left(\hat{\mathbf{X}}_{t_n}^{(k)} - \mathcal{H}\left(t_n, t_{n-1}, \hat{\mathbf{X}}_{t_{n-1}}^{(k)}, \hat{\mathbf{X}}_{t_N}(\hat{\mathbf{X}}_{t_{n-1}}^{(k)}, \theta)\right)\right) \\
&\overset{(a)}{=} \hat{\mathbf{X}}_{t_n}^{(k)} - \mathcal{G}^{(k)}\left(\hat{\mathbf{X}}_{t_n}^{(k)} - \mathcal{H}\left(t_n, t_{n-1}, \mathbf{X}_{t_{n-1}}, \hat{\mathbf{X}}_{t_N}(\mathbf{X}_{t_{n-1}}, \theta)\right)\right) \\
&\overset{(b)}{=} \mathcal{H}\left(t_n, t_{n-1}, \mathbf{X}_{t_{n-1}}, \hat{\mathbf{X}}_{t_N}(\mathbf{X}_{t_{n-1}}, \theta)\right) \\
&= \mathbf{X}_{t_n}
\end{aligned} \tag{45}$$

where:

- Eq. (a) follows from the induction hypothesis that $\hat{\mathbf{X}}_{t_{n-1}}^{(k)} = \mathbf{X}_{t_{n-1}}$

- Eq. (b) holds since $\mathcal{G}^{(k)} = \mathbf{I}$ or $\mathcal{G}^{(k)} = \mathcal{J}^{-1} = \mathbf{I}$ where $\mathcal{J} = \frac{\partial}{\partial \hat{\mathbf{X}}_{t_n}^{(k)}}\left(\hat{\mathbf{X}}_{t_n}^{(k)} - \mathcal{H}\left(t_n, t_{n-1}, \mathbf{X}_{t_{n-1}}, \hat{\mathbf{X}}_{t_N}\left(\mathbf{X}_{t_{n-1}}, \theta\right)\right)\right) = \mathbf{I}$

In the worst case, convergence is achieved in $K = N$ iterations as the convergence front must advances exactly one step per iteration. In practice, due to the immediate propagation of updated components, convergence is typically achieved in $K \ll N$ iterations.

