# OpenReview forum: "ParaSolver-Turbo: Accelerating Parallel Diffusion Integrator via Intrinsic Partially Linear Structure"
_ICLR.cc/2026/Conference — Submitted to ICLR 2026_

### Official Review · Reviewer_N49d · 2025-10-28

**Soundness:** 3
**Presentation:** 2
**Contribution:** 3
**Rating:** 4
**Confidence:** 3

**Summary:**

This paper considers accelerating the diffusion sampling process by solving a set of nonlinear and linear equations in a parallel manner. A so-called ParaSolver-Turbo method is proposed as an extension of ParalSolver. A sliding window approach is proposed to turn certain linear equations into nonlinear equations to improve sampling quality. Experimental results indicates the fast sampling process in terms of wall-clock in comparison to other parallel methods.

**Strengths:**

The main contribution of the paper is to introduce both a set of linear and nonlinear equations in the sampling process, where the set of linear equations sit at the low noise region (closer to the estimated clean image) while the set of nonlinear equations sit at the high noise region (far away from the estimated clean image). Basically, the set of linear equations are obtained by using a common estimated clean image.  I think it is because of the introduction of linear equations, which makes it more computationally efficient than other parallel sampling methods.

**Weaknesses:**

(1) I don't think Proposition 2 is correct. That is, for different values of ϱ, the ϱ-nonlinear equation system
in Eq. (10) is NOT equivalent. This is because the assumption that the diffusion model has a perfect noise predictor does not hold in practice. I don't think it is a widely used assumption. If the diffusion model has a perfect noise predictor, 1-step sampling would be enough. If the assumption hold, the conclusion in Proposition 2 does not provide any practical guidance.

(2) In the paragraph of "Hyperparameter Settings" in Section 6, the authors should mention how the hyper-parameters are set for ParaSolver-Turbo, rather than saying "More details are provided in the Appendix." I would think the setup for the parameter rho is crucial.

(3) In Table 1, one thing I don't understand is why for 1000 steps, the NFE needed for the new method much smaller than that of DDPM. On the other hand, for steps of 25 and 50, the NFE for the new method is larger than that of DDPM. Is it because for 1000 steps, the new method performs sampling over a set of coarse timesteps? If it is the case, then it is not a fair comparison w.r.t. the ParaSolver.

**Questions:**

(1) Right below Equ. (4), how come N=T where N is timestep index?

(2)  The authors only state "More details are provided in the Appendix." in a few places without specifying the section in appendix.

(3) Please specify the number of GPUs being used in Table 1 for each method.

---

> ### Author Response · Authors · 2025-11-23
>
> We thank Reviewer N49d for correctly identifying our core contribution and for the detailed, critical questions.
>
> > *I don't think Proposition 2 is correct. That is, for different values of ϱ, the ϱ-nonlinear equation system in Eq. (10) is NOT equivalent. This is because the assumption that the diffusion model has a perfect noise predictor does not hold in practice.*
>
> We clarify that **Prop. 2 is correct even when using an imperfect denoising model. In the revised version, we have removed the assumption of a perfect noise predictor and updated Proposition 1 and 2 to reflect realistic conditions where the denoising model contains estimation errors.** Our analysis demonstrates that this error is effectively suppressed by a coefficient near 0 throughout the sampling process, resulting in a negligible error term. This ensures the $\varrho$-nonlinear equation system  remains  equivalent to the true sequential integral trajectory for any different values of $\varrho$.
>
> **We are really excited about these significant findings**  as they bring several interesting and helpful findings. For example, Proposition 1 reveals that the exact sequential solver possesses an intrinsically simpler linear structure. We believe this linear structure can benefit the community significantly. Theoretically, it bridges the gap between complex non-linear dynamics and well-behaved linear systems. Engineering-wise, this discovery implies broad potential: it not only unlocks parallel sampling to minimize latency, but also may simplify the inversion process into a stable linear problem for precise image editing, and provide tighter error bounds to guarantee the convergence of efficient few-step solvers.
>
> **We warmly invite you to see the general response and Proposition 1 and 2 in the revised manuscript for further details.**
>
>
> > *In the paragraph of "Hyperparameter Settings" in Section 6, the authors should mention how the hyper-parameters are set for ParaSolver-Turbo, rather than saying "More details are provided in the Appendix." I would think the setup for the parameter $\varrho$ is crucial.*
>
>  We set $\varrho=8$ for ParaSolver-Turbo, as we utilize 8 NVIDIA 3090 GPUs. We have updated this on the  manuscript.
>
> > *In Table 1, one thing I don't understand is why for 1000 steps, the NFE needed for the new method much smaller than that of DDPM. On the other hand, for steps of 25 and 50, the NFE for the new method is larger than that of DDPM. Is it because for 1000 steps, the new method performs sampling over a set of coarse timesteps? If it is the case, then it is not a fair comparison w.r.t. the ParaSolver.*
>
> We clarify that **the comparison is completely fair**. All methods, including our ParaSolver-Turbo, operate on the exact same set of discretized timesteps (i.e., the time schedule $N$ is identical for all baselines and our method). Specifically: For the 1000-step experiment, we set $N=1000$ for all methods. For the 50-step and 25-step experiments, we set $N=50$ and $N=25$ for all methods, respectively.
>
> The reduction in NFE at 1000 steps arises because the efficiency gains from the linear solver (which skips NFEs for the vast majority of steps) strictly outweigh the overhead of parallel iterations ($K \ll N$). Conversely, at few steps (e.g., $N=25$), the parallel iteration count $K$ is comparable to $N$, so the cumulative NFE during these iterations exceeds the single-pass cost of sequential solvers.
>
> > *Right below Equ. (4), how come N=T where N is timestep index?*
>
> We  divides  $T$ discrete time steps into $N$ sub-intervals $[t_0,t_1,\ldots,t_N]$ with $0=t_0<t_1<\cdots<t_N=T$. Existing parallel methods like ParaDiGMS take $t_0=0,t_1=1,\cdots,t_N=T$, therefore $N$ is equal to $T$.
>
>
> > *The authors only state "More details are provided in the Appendix." in a few places without specifying the section in appendix.*
>
>   Thanks for the attention to detail! We have specified the section in appendix.
>
> > *Please specify the number of GPUs being used in Table 1 for each method.*
>
> All experiments in Table 1 were conducted using 8 NVIDIA RTX 3090 GPUs, as stated in Appendix D (Lines 865-866).

---

> ### Author Response · Authors · 2025-11-25
> **Looking forward to your further comments**
>
> Dear Reviewer **N49d**
>
> Thank you for taking the time to review our manuscript and for your valuable feedback and recognition. We have carefully addressed all the comments and concerns raised, as reflected in our detailed responses and the revised manuscript and supplementary material.
>
> We sincerely appreciate your efforts and look forward to your further assessment.
>
> Best regards,
>
> The Authors

---

> > ### Author Response · Authors · 2025-11-26
> > **Looking forward to your further feedback**
> >
> > Dear Reviewer **N49d**,
> >
> > Have our comprehensive responses address your concerns? We are looking forward to your further feedback.
> >
> > Best regards,
> >
> > The authors

---

### Official Review · Reviewer_qrk9 · 2025-11-01

**Soundness:** 3
**Presentation:** 2
**Contribution:** 3
**Rating:** 6
**Confidence:** 4

**Summary:**

This work studies how to accelerate the reverse process of diffusion models by designing a parallel sampling algorithm, proposes the ParaSolver-Turbo algorithm, and achieves a 71.4x speedup. To achieve this goal, this work proposes the $\varrho$-nonlinear equation system, which combines the nonlinear score and linear conditional score, to avoid a large NFEs.

**Strengths:**

1.	From the empirical perspective, ParaSolver-Turbo achieves better performance compared with DDPM, DDIM, DPMsolver, and ParaSolver, ParaDiGMs, with a much faster speed.

**Weaknesses:**

A minor concern is the assumption. In the proof of Proposition 1, Eq. (21) uses the analytic form of the condition score $\nabla \log p(X_t|X_T)$ for the unconditional score $\nabla \log p(X_t)$, which is also adopted by Song et al. It would be better to discuss this assumption in the main content.Questions:

**Questions:**

Please see my weakness.

1.	Since the flow-based models is popular, it would be better to do experiments in the flow-based models (such as flux or SD 3.5) to show the advantage of the algorithm.

Comments:

1.	It would be better to rewrite lines 712.

2.	It would be better to add the discussion of limitations and broader impacts (in the appendix if there is no space in the main content).

---

> ### Author Response · Authors · 2025-11-23
>
> We thank Reviewer qrk9 for the positive assessment of our empirical results.
>
> > *A minor concern is the assumption. In the proof of Proposition 1, Eq. (21) uses the analytic form of the condition score $\nabla \log p(X_t|X_T)$ for the unconditional score $\nabla \log p(X_t)$, which is also adopted by Song et al. It would be better to discuss this assumption in the main content.*
>
> Thank you for your question! In the revised version, we have removed such an assumption and updated Propositions 1 and 2 to reflect realistic conditions where the denoising model contains estimation errors.
>
> > *Since the flow-based models are popular, it would be better to do experiments in the flow-based models (such as flux or SD 3.5) to show the advantage of the algorithm.*
>
> Thank you for this important suggestion. This is a very interesting research direction. In view of this, we implement a demo of our method on the flow-based model Flux, where the nonlinear degree in our method is set as the number of sequential steps.
>
> The results demonstrate that our method delivers significant acceleration on Flux. As shown in the table below, at $100$ steps, it reduces the required iterations to $23.49$, yielding a $4.0\times$ speedup ($87.94$s vs. $22.16$s) while achieving comparable  FID and CLIP scores. The acceleration remains consistent across different step counts, reaching $1.8\times$ even at $25$ steps, confirming the effectiveness of our method as a general-purpose accelerator.
>
> Table 1 **Quantitative comparisons on 12B Flux model ($1024\times1024$ resolution, 10k random samples from ImageNet for FID and CLIP computation)**, the tolerance is $0.01$ and the nonlinear degree is set as the number of sequential steps.
>
> | Steps | Method | Iters $\downarrow$  | CLIP $\uparrow$ | FID $\downarrow$ | Time (s) $\downarrow$ | Speedup $\uparrow$ |
> | :---: | :---| :---: | :---: | :---: | :---: | :---: |
> | **100** | Euler | 100.00 |  32.31 | 55.62 | 87.94 | 1.0$\times$ |
> | | Euler + ours | **23.49** |  32.40 | 55.28 | **22.16** | **4.0$\times$** |
> | **75** | Euler | 75.00 |  32.36 | 55.25 | 66.15 | 1.0$\times$ |
> | | Euler + ours | **19.32** | 32.42 | 55.35 | **18.38** | **3.6$\times$** |
> | **50** | Euler | 50.00 |  32.38 | 54.89 | 44.38 | 1.0$\times$ |
> | | Euler + ours | **15.11** |  32.45 | 55.84 | **14.57** | **3.0$\times$** |
> | **25** | Euler | 25.00 |  32.40 | 55.23 | 22.60 | 1.0$\times$ |
> | | Euler + ours | **12.79** |  32.47 | 55.88 | **12.42** | **1.8$\times$** |
>
> > *It would be better to rewrite lines 712.*
>
> This line is a table of contents: sub-chapter titles. The revised version has shortened this entry.
>
> > *It would be better to add the discussion of limitations and broader impacts (in the appendix if there is no space in the main content).*
>
> We clarify that we have already included a detailed **"Limitations, Future Directions, and Broader Impacts"** section in **Appendix B**.
>
>
> We hope this resolves your concerns. It is heartening to hear that you praise "good soundness and contribution". We are looking forward to your response.

---

> ### Author Response · Authors · 2025-11-25
> **Thanks for your feedback**
>
> Dear Reviewer **qrk9**
>
> Thank you for recognizing the value of our work. We sincerely appreciate your thoughtful review and the decision to raise your rating.
>
> Best regards,
>
> The authors

---

### Official Review · Reviewer_xK8W · 2025-11-01

**Soundness:** 3
**Presentation:** 2
**Contribution:** 2
**Rating:** 4
**Confidence:** 3

**Summary:**

This paper proposes ParaSolver-Turbo, a method that formulates diffusion sampling as solving a system of nonlinear equations and accelerates it through parallel sampling. Building upon previous works, it reformulates the diffusion differential equation, which is traditionally solved sequentially, into a system of banded nonlinear equations. The paper then introduces a linearized version of this system by incorporating the final clean sample estimated by the neural network. By combining the linear and nonlinear systems, the proposed method strikes a better balance between computational cost and solution accuracy. Experiments demonstrate that ParaSolver-Turbo achieves greater speedup than baseline methods such as ParaDiGMS and ParaSolver.

**Strengths:**

1. Combining parallel sampling and diffusion solver is a relatively new idea.
2. Introducing a linear system to reduce computational cost sounds a reasonable method.

**Weaknesses:**

1. My major concern lies in the experimental results and settings.
    - As shown in the main table, compared with traditional solvers, the iteration compression ratio of ParaSolver-Turbo is generally smaller than its speedup ratio (e.g., for DDIM with 25 steps, the iteration compression ratio is 25/11=2.27<3.1). However, in my understanding, since parallel computation increases the computation amount per iteration, the speedup should not exceed the iteration compression ratio. Could the authors provide explanation to this?
    - From my experience, image generation is typically a computation-intensive task, where the actual computation cost dominates the inference time rather than the number of model calls, especially for large models like Stable Diffusion. Therefore, the actual speedup ratio should correlate more with NFE than with iterations. Achieving a speedup through parallel sampling seems strange to me. \textbf{I suggest reporting the batch size per GPU and GPU utilization, and adding comparison results with fewer-step solvers in Table 2.}
        - Moreover, for video generation tasks, diffusion models contribute an even larger portion of the computation cost, so this approach appears to have limited scalability for such larger tasks.
    - Some of the baselines compared in this paper are somewhat outdated. Current traditional sequential solvers have long achieved 5–10 step sampling [1–4], and the authors should discuss or compare the proposed method with these methods. Some of the claims about speedup are still made against the earliest 1000-step DDPM, which seems overclaimed.

2. I suggest the authors add a more detailed discussion on ParaSolver-Turbo in relation to previous parallel sampling methods, such as ParaSolver, ParaDiGMS, and ParaTAA. This would help readers quickly grasp the landscape of this research area and understand the proposed method.

[1] Zheng, K., Lu, C., Chen, J., & Zhu, J. (2023). Dpm-solver-v3: Improved diffusion ode solver with empirical model statistics. Advances in Neural Information Processing Systems, 36, 55502-55542.

[2] Zhou, Z., Chen, D., Wang, C., & Chen, C. (2024). Fast ode-based sampling for diffusion models in around 5 steps. In Proceedings of the IEEE/CVF Conference on Computer Vision and Pattern Recognition (pp. 7777-7786).

[3] Liu, E., Ning, X., Yang, H., & Wang, Y. (2024). A unified sampling framework for solver searching of diffusion probabilistic models. In The Twelfth International Conference on Learning Representations.

[4] Liang, Y., Fang, X., Chen, H., & Wang, Y. Linear Multistep Solver Distillation for Fast Sampling of Diffusion Models. In The Thirteenth International Conference on Learning Representations.

**Questions:**

1. Since the linear system is introduced to reduce the number of denoised samples per step, which introduces additional errors (because the predicted clean sample is only an expectation and is different from the true clean sample), how can ParaSolver-Turbo achieve fewer iterations in some cases compared to ParaSolver?

---

> ### Author Response · Authors · 2025-11-23
>
> We thank Reviewer xK8W for the insightful comments on our experimental results and setup.
>
> >  *Since parallel computation increases the computation amount per iteration, the speedup should not exceed the iteration compression ratio. Achieving a speedup through parallel sampling seems strange to me.*
>
> It is intuitive to think that "since parallel algorithms increase the computational load per iteration, the speedup should not exceed the iteration compression ratio".
>
> **However, there is a critical misconception**: while parallel algorithms do typically increase the total computational load (e.g., total FLOPs) per iteration,  this does not necessarily translate to increased wall-clock time per iteration.  We clarify that **the time for a parallel iteration can often be faster than the time for a single sequential step, which is well-documented in the literature [1, 2, 3].  This further leads to the speedup exceeding the iteration compression ratio.** E.g., FlashAttention[1] shows that "Even with the increased FLOPs, our algorithm runs faster (7.6x on GPT-2) than standard attention, thanks to the massively reduced amount of HBM access."
>
> The less time comes from two aspects.
>
> First,  parallelism distributes the increasing computational load across N computing units, so that each unit only needs to handle 1/N of the workload, such as data parallelism. Besides, our method utilizes a partial linear structure, dramatically lowering the computational load, thereby further decreasing wall-clock time per iteration.
>
> Second, parallelism  consolidates high-latency system overheads like CUDA kernel launches into a single iteration, leading to significant decreases in the system overhead. Taking the 25-step iteration as an example, the sequential baseline must execute a Python loop 25 times. Each loop incurs significant CPU-side system overheads, including the Python GIL lock, scheduler coefficient calculations, moving tensors to the GPU, and triggering CUDA kernel launches.   In contrast, these overheads are packed into only 11 iterations by our method.
>
> [1] FlashAttention: Fast and Memory-Efficient Exact Attention with IO-Awareness
>
> [2] Blockwise Parallel Decoding for Deep Autoregressive Models
>
> [3] ZeRO: Memory Optimizations Toward Training Trillion Parameter Models
>
> > *The batch size per GPU and GPU utilization, and adding results with fewer-step solvers in Tab. 2.*
>
> We have added these information to Tab. 2.
>
> >  *Limited scalability for video generation tasks.*
>
> **We respectfully disagree with this**. First, our method requires each GPU to maintain only one model copy. Second, the partial linear design reduces noisy samples to just the number of GPUs, enabling entirely feasible parallel execution.  We believe our linear design is highly suitable for video generation tasks.
>
> > *Compare with solvers in 5–10 step sampling.*
>
> We respectfully argue that directly comparing our speedup against 5-10 step' solvers is unfair, as they suffer from noticeable quality loss compared to 20-50 step. In contrast, our **ParaSolver-Turbo achieves few-step latency without the loss of quality**.
>
> To this end, we experimented on CIFAR-10 using DPM-Solver-v3. As shown, the 10-step setting degrades FID significantly (2.64 to 3.40). However, our method reduces the 25-step solver to 10 steps, achieving a comparable speedup (2.54x) while maintaining a superior FID (2.58).
>
> |Method|Iters|FID $\downarrow$ |Speedup|
> |-|-|-|-|
> | DPM-Solver-v3 | 25  | 2.64 | 1.00x
> | DPM-Solver-v3 | 10  | 3.40 | 2.58x
> | Ours+DPM-Solver-v3 |10| **2.58** | 2.54x
>
> > *More detailed discussion on  ParaSolver, ParaDiGMS, and ParaTAA.*
>
> We clarify that both Lines 83-97 and Lines 239-241 have discussed these methods in detail.
>
> > *Since the linear system introduces additional errors, how can ParaSolver-Turbo achieve fewer iterations compared to ParaSolver?*
>
>
> We clarify that ParaSolver-Turbo only aims to reduce the number of denoised samples per step rather than achieve fewer iterations.
>
> As for the error, we have some exciting theoretical results. Our  Proposition 1&2 demonstrates that the error is effectively suppressed by a coefficient near 0 throughout the sampling process, resulting in a negligible error term. This ensures the linear equation system remains equivalent to the true sequential integral trajectory.
>
> **We are really excited about these significant findings**  as they bring several interesting and helpful findings. For example, Prop. 1 reveals that the exact sequential solver possesses an intrinsically simpler linear structure. We believe this linear structure can benefit the community significantly. It may simplify the inversion process into a stable linear problem for precise image editing and provide tighter error bounds to guarantee the convergence of efficient few-step solvers.
>
> Thank you for guiding us toward this important improvement! **We warmly invite you to see the general response and the revised Propositions 1&2  for further details.**

---

> ### Author Response · Authors · 2025-11-25
> **Looking forward to your further comments**
>
> Dear Reviewer **xK8W**,
>
> Thank you for taking the time to review our manuscript and for your valuable feedback and recognition. We have carefully addressed all the comments and concerns raised, as reflected in our detailed responses and the revised manuscript and supplementary material.
>
> We sincerely appreciate your efforts and look forward to your further assessment.
>
> Best regards,
>
> The Authors

---

> > ### Author Response · Authors · 2025-11-26
> > **Looking forward to your further feedback**
> >
> > Dear Reviewer **xK8W**,
> >
> > Have our comprehensive responses address your concerns? We are looking forward to your further feedback.
> >
> > Best regards,
> >
> > The authors

---

### Official Review · Reviewer_fMT2 · 2025-11-01

**Soundness:** 3
**Presentation:** 3
**Contribution:** 3
**Rating:** 6
**Confidence:** 3

**Summary:**

This paper addresses the challenge of accelerating the inference of diffusion models. The authors proposed ParaSolver-Turbo, a parallel sampling framework that exploits a partial linear structure in the diffusion integral formulation. The experimental result show that the proposed solver outperforms previous parallel sampling methods in inference time while maintaining generation quality.

**Strengths:**

1. The reformulation of sampling process into a partially linear system is novel.
2. Theoretical results are provided, assuming an ideal denoiser.
3. ParaSolver-Turbo has good empirical performance. It is faster than existing parallel sampling methods while maintaining similar FID and CLIP scores.

**Weaknesses:**

1. Both the reformulation and theoretical analysis rely critically on the ideal denoising assumption, which is theoretically impossible. This weakens the theoretical basis of the proposed method and undermines the claimed theoretical properties in realistic settings.

2. The paper mentions that one-step methods like diffusion distillation and consistency models reduce sample quality, but the experiments do not compare ParaSolver-Turbo with these methods. It is unclear whether the proposed approach achieves better quality.

**Questions:**

1. In the proof of Proposition 1, what's the justification for the first equality in (21)? It is insufficient to say following Song et al 2023b.

2. Lines 206-208, why does it need information regarding the target data distribution if it equivalent as claimed?

3. What's the implication of the failure of the ideal denoising assumption?

4. Is there any rule for choosing $\rho$?

---

> ### Author Response · Authors · 2025-11-23
>
> We thank Reviewer fMT2 for the positive feedback on our method's novelty and empirical performance.
>
> > *Both the reformulation and theoretical analysis rely critically on the ideal denoising assumption, which is theoretically impossible. This weakens the theoretical basis of the proposed method and undermines the claimed theoretical properties in realistic settings.*
>
>
>  **In the revised version, we have removed the assumption of a perfect noise predictor and updated Propositions 1 and 2 to reflect realistic conditions where the denoising model contains estimation errors.** Our analysis demonstrates that this error is effectively suppressed by a coefficient near 0 throughout the sampling process, resulting in a negligible error term.
>
> **We are really excited about these significant findings**  as this brings several interesting and helpful findings. For example, Proposition 1 reveals that the exact sequential solver possesses an intrinsically simpler linear structure. We believe this linear structure can benefit the community significantly. Theoretically, it bridges the gap between complex non-linear dynamics and well-behaved linear systems. Engineering-wise, this discovery implies broad potential: it not only unlocks parallel sampling to minimize latency, but also may simplify the inversion process into a stable linear problem for precise image editing, and provide tighter error bounds to guarantee the convergence of efficient few-step solvers.
>
> Thank you for guiding us toward this important improvement! **We warmly invite you to see the general response and Propositions 1 and 2 in the revised manuscript for further details.**
>
> > *The paper mentions that one-step methods like diffusion distillation and consistency models reduce sample quality, but the experiments do not compare ParaSolver-Turbo with these methods. It is unclear whether the proposed approach achieves better quality.*
>
> This is a fair point. We would like to clarify that ParaSolver-Turbo is a **training-free solver**  designed to accelerate any pre-trained diffusion model. In contrast, distillation and consistency models require a separate, and often costly, training phase; our method is thus orthogonal and complementary.
>
> Regarding sample quality, there is a broad consensus in the community that distillation and consistency models often exhibit lower generation quality compared to traditional sequential sampling methods like DDPM, due to the inherent trade-off between speed and fidelity [1,2,3]. In contrast, parallel sampling methods based on equation solving have been rigorously validated, both theoretically and empirically, to yield samples of equivalent quality to traditional sequential sampling [4]. Consequently, the generation quality of distillation and consistency models is inherently inferior to that of our parallel sampling approach.
>
>
> [1] Progressive Distillation for Fast Sampling of Diffusion Models
>
> [2] Consistency Models
>
> [3] Multistep Consistency Models
>
> [4] Parallel Sampling of Diffusion Models
>
>
> > *In the proof of Proposition 1, what's the justification for the first equality in (21)? It is insufficient to say following Song et al 2023b.*
>
> Thank you for this question. The first equality in Eq. (21) follows directly from Lemma 1 in Appendix A.3 of Song et al.'s Consistency Models paper.
>
>
>
> > *Lines 206-208, why does it need information regarding the target data distribution if it is equivalent as claimed?*
>
> Our Proposition 1 establishes the sequential integral solver as a linear solver that requires information about the target data samples. Consequently, without any information regarding the target data distribution, it is impossible to generate a sample from this distribution using the linear solver.
>
> Our method addresses this problem by utilizing the score model's predicted clean sample as input to the linear solver. This injects the learned data distribution into the solver without requiring external prior knowledge, thereby transforming it into an effective system.
>
> > *What's the implication of the failure of the ideal denoising assumption?*
>
> This is an excellent question that  alerted us to this problematic  assumption. In practice, this assumption *always* fails. The implication is:
>
> 1. The predicted clean sample is only an *estimate*, not the ground truth $X_T$.
>
> 2. This estimate introduces errors into the linear part of our solver.
>
> Given these,  we thus  removed the ideal denoising assumption and  updated Proposition 1 and 2 to reflect realistic conditions where the denoising model contains estimation errors.
>
> > *Is there any rule for choosing $\varrho$?*
>
>  As shown in our **Remark 5**, $\varrho$ is less of a hyperparameter and more of a hardware configuration parameter. For good efficiency, $\varrho$ can simply be set to the number of available GPUs.

---

> > ### Author Response · Authors · 2025-11-25
> > **Looking forward to your further comments.**
> >
> > Dear Reviewer **fMT2**,
> >
> > Thank you for taking the time to review our manuscript and for your valuable feedback and recognition. We have carefully addressed all the comments and concerns raised, as reflected in our detailed responses and the revised manuscript and supplementary material.
> >
> > We sincerely appreciate your efforts and look forward to your further assessment.
> >
> > Best regards,
> >
> > The Authors

---

> > > ### Author Response · Authors · 2025-11-26
> > > **Looking forward to your further feedback**
> > >
> > > Dear Reviewer **fMT2**,
> > >
> > > Have our comprehensive responses address your concerns? We are looking forward to your further feedback.
> > >
> > > Best regards,
> > >
> > > The authors

---

### Author Response · Authors · 2025-11-23
**General Response**

# General Response

We sincerely thank the reviewers for their constructive feedback and for acknowledging the value of our work. We are encouraged by the positive assessment on the **novel reformulation of the sampling process** (Reviewer fMT2), the **strong empirical performance** (Reviewers fMT2 and qrk9), and the **innovative linear and nonlinear equations** (Reviewers xK8W and N49d).

The reviewers raised a common and critical concern regarding the **validity of the ideal denoising assumption** (Reviewers fMT2, qrk9, and N49d). In response, we have deeply revised our theoretical framework and are thrilled to present some **exciting theoretical and empirical updates**:

* **Removal of Ideal Assumptions:** We have successfully removed the assumption of a perfect noise predictor. In the revised manuscript, we update **Proposition 1 and Proposition 2** to reflect realistic conditions where the denoising model contains estimation errors. Our new analysis demonstrates that **this error is effectively suppressed by a coefficient near 0 throughout the sampling process**, ensuring the errors remain negligible.
* **Discovery of Intrinsic Linear Structure:** This theoretical update reveals a significant finding: **the exact sequential solver possesses an intrinsically simpler linear structure.** This bridges the gap between complex non-linear dynamics and well-behaved linear systems, offering broad potential for future tasks such as image editing inversion and tighter error bound guarantees.
* **Generalization to Flow-Based Models (Flux):** To demonstrate the generality of our approach (suggested by Reviewer qrk9), we applied ParaSolver-Turbo to the **12B parameter Flux model**. We are excited to report that our method achieves a **4.0$\times$ speedup** while maintaining comparable FID and CLIP scores, proving its effectiveness beyond standard diffusion models.

**We sincerely invite you to refer to the updated Propositions 1 & 2 and the new Flux experiments.** We are genuinely excited about these findings, as they not only directly address the reviewers' concerns but also provide new inspiration for utilizing linear structures in generative modeling.

**Last but not least**, we thank the PCs, ACs, and reviewers again for their invaluable time. **We commit that the source code necessary for conducting the experiments will be made publicly available upon publication**. We will ensuring that the final accepted manuscript includes a link to the accessible code, along with the newly added experiments and analyses.

---

### Meta-Review · Area_Chair_4MPD · 2026-01-06

**Summary:**

The paper introduces "ParaSolver-Turbo," a parallel sampling algorithm designed to accelerate the inference of Diffusion Probabilistic Models (DPMs). The method builds upon the ParaSolver framework by identifying and exploiting an intrinsic partial linear structure within the diffusion trajectory. By reformulating the sequential integral solver into a system combining linear and non-linear equations, the authors aim to achieve faster convergence in a parallel setting.

**Reviewer Concerns:**

Addressed Concerns:
- Experimental Scope: The authors successfully addressed Reviewer qrk9's request for validation on modern flow-based models (e.g., Flux), demonstrating a ~4x speedup, which strengthens the empirical evaluation.
- Clarification on Speedup: The authors provided a reasonable explanation to Reviewer xK8W regarding how wall-clock speedup can exceed iteration compression ratios due to the reduction of system overheads (e.g., Python GIL, kernel launches) in parallel execution.

Outstanding Concerns:
- Theoretical Soundness (Critical): A major concern raised by Reviewers fMT2, qrk9, and N49d was the reliance on an "ideal denoising" assumption (perfect score estimation). In the rebuttal, the authors claimed to remove this assumption by introducing an error term ($\zeta$) and arguing it is negligible. However, I find this resolution unsatisfactory for several reasons:
  1. While the authors argue the error influence is near zero, this holds primarily for Variance Preserving (VP) diffusion models. In Variance Exploding (VE) models, this error term is not necessarily negligible and can be significant. The theoretical "fix" does not generalize as claimed.
  2. The critical assumptions and the definition of the error term (and key notations like $\alpha_t$) are relegated to the appendix. The main text remains theoretically incomplete without these definitions, which obscures the limitations of the proposed theorems.
- Overstated Claims on Speedup: The claimed "40x-50x" speedup is achieved only under the specific and less practical setting of compressing 1000 steps. In more standard regimes (25-50 steps), the speedup is modest (approx. 2x-4x), which reviewers noted as a discrepancy between the abstract's claims and practical utility.

**Reviewer Scores:**

- Reviewer qrk9 (6): Would likely lower or remain their score. This reviewer raised the initial concern about the ideal assumption but accepted the rebuttal's "fix." Had the distinction between VP and VE error terms been highlighted during the discussion, they would likely have recognized the theoretical gap.
- Reviewer fMT2 (6): Would likely lower or remain their score. This reviewer correctly identified the theoretical weakness initially ("weakens the theoretical basis"), and the rebuttal effectively moved the assumption to the appendix rather than solving it rigorously.
- Reviewer xK8W (4): Score would likely remain unchanged .
- Reviewer N49d (4): Score would likely remain unchanged .

---

### Decision · Program_Chairs · 2026-01-26

Reject